# Cryo-ET detects bundled triple helices but not ladders in meiotic budding yeast

Olivia X. Ma[1¤], Wen Guan Chong[1], Joy K. E. Lee[1], Shujun Cai[1], C. Alistair Siebert[2], Andrew Howe[2], Peijun Zhang[2,3], Jian Shi[1], Uttam Surana[4,5,6,7], Lu Gan[1]*

1 Department of Biological Sciences and Centre for BioImaging Sciences, National University of Singapore, Singapore, Singapore, 2 Diamond Light Source Ltd, Harwell Science & Innovation Campus, Didcot, Oxfordshire, United Kingdom, 3 Division of Structural Biology, Wellcome Centre for Human Genetics, University of Oxford, Oxford, United Kingdom, 4 Institute of Molecular and Cell Biology, Agency for Science, Technology and Research (A*STAR), Proteos, Singapore, 5 Bioprocessing Technology Institute, A*STAR, Singapore, Singapore, 6 Biotransformation Innovation Platform, A*STAR, Singapore, Singapore, 7 Department of Pharmacology, National University of Singapore, Singapore, Singapore

¤ Current address: Southern University of Science and Technology, Shenzhen, People's Republic of China
* lu@anaphase.org

**Data Availability Statement:** The subtomogram averages are available at EMDB under accession code EMD-31081. The cryo-ET raw data are available at EMPIAR under accession code EMPIAR-10670. The raw confocal microscopy data

## Abstract

In meiosis, cells undergo two sequential rounds of cell division, termed meiosis I and meiosis II. Textbook models of the meiosis I substage called pachytene show that nuclei have conspicuous 100-nm-wide, ladder-like synaptonemal complexes and ordered chromatin loops. It remains unknown if these cells have any other large, meiosis-related intranuclear structures. Here we present cryo-ET analysis of frozen-hydrated budding yeast cells before, during, and after pachytene. We found no cryo-ET densities that resemble dense ladder-like structures or ordered chromatin loops. Instead, we found large numbers of 12-nm-wide triple-helices that pack into ordered bundles. These structures, herein called meiotic triple helices (MTHs), are present in meiotic cells, but not in interphase cells. MTHs are enriched in the nucleus but not enriched in the cytoplasm. Bundles of MTHs form at the same timeframe as synaptonemal complexes (SCs) in wild-type cells and in mutant cells that are unable to form SCs. These results suggest that in yeast, SCs coexist with previously unreported large, ordered assemblies.

## Introduction

Meiosis is a conserved form of cell division in which diploid cells undergo one round of DNA replication followed by two sequential rounds of cell division. Each of the resulting gametes contains half the number of parental chromosomes. Homologous chromosomes segregate in the first division (meiosis I) and sister chromatids segregate in the second division (meiosis II). While meiosis II closely resembles mitosis (vegetative proliferation) with respect to chromosome segregation, meiosis I displays very different chromosome behaviors. For instance, the chromosomes in early meiosis I cells undergo rapid cytoplasmic actin-dependent motions [1] and acquire programmed double-strand breaks [2]. Subsequently, the homologous

are available at the BioImage Archive as entry S-BIAD293.

**Funding:** Ministry of Education – Singapore (MOE) R-154-000-A49-114: Dr Olivia X. Ma, Wen Guan Chong, Joy K.E. Lee, Dr Shujun Cai, Dr Lu Gan; Ministry of Education – Singapore (MOE) R-154-000-B42-114: Dr Olivia X. Ma, Wen Guan Chong, Joy K.E. Lee, Dr Shujun Cai, Dr Lu Gan; Ministry of Education – Singapore (MOE) MOE2019-T2-2-045: Dr Olivia X. Ma, Wen Guan Chong, Joy K.E. Lee, Dr Shujun Cai, Dr Lu Gan; A*STAR | Biomedical Research Council (BMRC): Dr Uttam Surana; BBRSC: Dr Alistair Siebert, Dr Andrew Howe, Dr Peijun Zhang; UKRI | Medical Research Council (MRC): Dr Alistair Siebert, Dr Andrew Howe, Dr Peijun Zhang; Wellcome Trust: Dr Alistair Siebert, Dr Andrew Howe, Dr Peijun Zhang. The funders had no role in study design, data collection and analysis, decision to publish, or preparation of the manuscript.

**Competing interests:** No, there is no conflict of interest. My manuscript contains the following statement: "The authors declare that they have no conflict of interest."

**Abbreviations:** MTH, meiotic triple helix; SC, synaptonemal complex; SM, Sporulation medium; cryo-EM, electron cryomicroscopy / cryo-electron microscopy; cryo-ET, electron cryotomography / cryo-electron tomography.

chromosomes align along their long axes and undergo recombination using machines that are not present in mitosis [3].

The budding yeast *Saccharomyces cerevisiae* (herein called yeast) uses conserved macromolecular machines to carry out chromosome alignment, homologous recombination, and segregation during meiosis I. The most iconic among the meiosis-related structures is the SC, a conserved 100-nm-wide protein assembly that forms end-to-end between homologous chromosomes [4, 5]. In traditional electron-microscopy images, the SC displays a ladder-like organization, comprised of two parallel lateral elements (the rails), which are bridged by densely packed rung-like features in the central region [6, 7]. The central region has at least two sub-components–the transverse filaments (the rungs), which align perpendicular to the lateral elements, and the central element, which aligns parallel to the lateral elements [8, 9]. Homologous chromosomes are believed to arrange as chromatin loops that anchor at and project from the lateral elements [10]. Despite the low sequence conservation of its component proteins, the SC's structural features are common to most eukaryotes [11–14].

Meiotic nuclei have been studied for decades by traditional EM [11, 12], but not by electron cryotomography (cryo-ET). Cryo-ET can reveal 3-D nanoscale structural details of cellular structures in a life-like state because the samples are kept unfixed, unstained, and frozen-hydrated during all stages of sample preparation and imaging [15]. The densities seen in cryo-ET data come from electron scattering of the biological macromolecules. In comparison, the densities seen in traditional EM are from electron scattering of heavy metals such as uranium, tungsten, and osmium, which have adhered to a subset of biological macromolecules that were not extracted in earlier steps. Here we use cryo-ET to visualize the organization of meiotic yeast nuclei throughout meiosis I, with specific focus on pachytene. We find that the pachytene nuclei lack ladder-like densities; instead, they are enriched in bundles of meiotic triple helices (MTHs). Time course experiments and chemical perturbations show that these MTHs coincide with the formation of SCs and are sensitive to 1,6-hexanediol and Latrunculin A. MTHs are still present in mutants that are unable to form SCs. Our analysis shows that MTHs coexist with SCs, which have an unknown cryo-ET structure.

## Results

To make the cryo-ET analysis of meiotic cells feasible, we used strains of the SK1 background [16]. SK1 cells that are grown first in pre-sporulation medium and then in sporulation medium (SM) will synchronously assemble SCs at 4 hours and then disassemble SCs after 7 hours in SM. We used both NKY611 and DK428 strains as wild type (WT) and we used EW104 (*ndt80Δ*) to enrich for pachytene cells [17–21]. Strains DK428 and EW104 bear GFP-tagged Zip1 at its native locus [20, 21] to track the assembly of the SCs, which are markers of pachytene. With few exceptions, we use the terms wild-type (WT) and *ndt80Δ* instead of the strain-number designations. To verify the synchronization, we switched WT cells from pre-sporulation medium to SM and then classified their Zip1-GFP fluorescence signals as a function of time based on published criteria [20] (Fig 1A). Diffuse Zip1-GFP fluorescence appeared in 85% of cells (n = 75) after 3 hours in SM (Fig 1B). The fluorescence signals appeared punctate at 4 hours in SM, string-like in 5–6 hours, started to weaken in 7 hours, and were rarely visible by 8 hours (Fig 1B). Meiosis completed after 12 hours of SM incubation, at which point 97% of the cells (n = 155) had an ascus with 4 spores. WT cultures incubated 5–6 hours in SM had the highest proportion of mature SC-containing cells (Fig 1C). In comparison, *ndt80Δ* cells started to accumulate intranuclear Zip1-GFP approximately 3 hours after SM incubation and then arrested in pachytene as expected (Fig 1D). All these results are consistent with previous studies [20, 21].

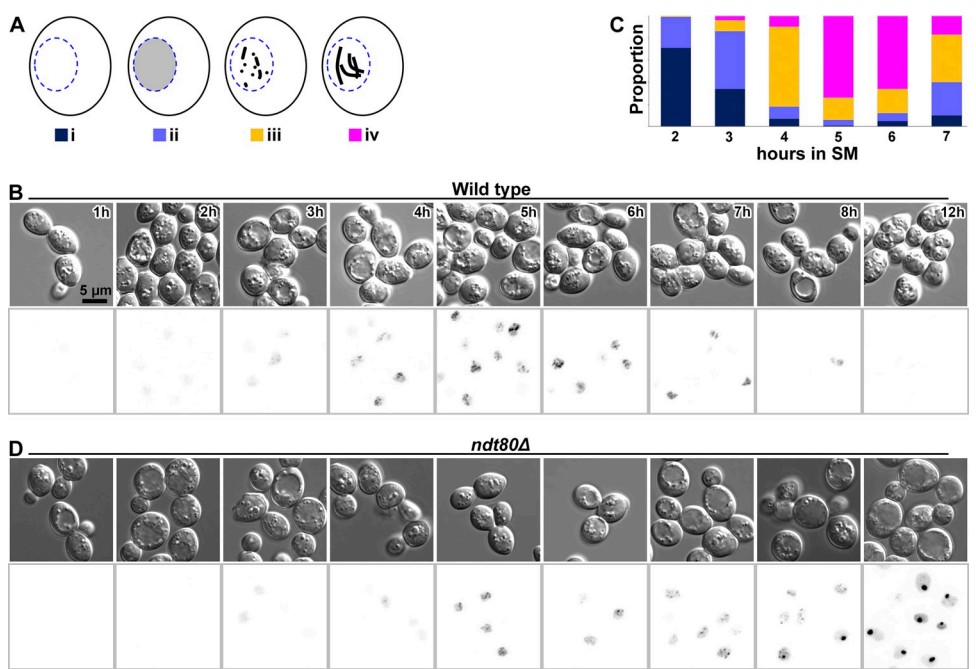

**Fig 1. Meiosis monitored by their SC assembly states.** (A) Cartoon of the different classes of Zip1-GFP fluorescence in meiotic prophase. i, non-fluorescent; ii, diffuse fluorescence; iii, punctate fluorescence; iv, string-like fluorescence corresponding to full-length SCs. The nucleus is blue and the Zip1-GFP signal is gray/black inside the nucleus. This schematic is based on White, et al. [20]. (B) Time course of WT cells expressing Zip1-GFP in SM. Diffuse fluorescence started to appear after 2 hours and became punctate between 3 and 4 hours. Long linear SCs appeared after 5 hours in SM and became rarer after 7 hours. The right-most brightfield image shows WT cells after 12 hours in SM. Each ascus contained four spores. Upper panels: differential interference contrast; lower panels: Zip1-GFP fluorescence in inverted contrast. (C) Time course for SC formation in WT cells based on the Zip1-GFP fluorescence pattern. Approximately 100 cells were sampled at each time point. The color scheme is indicated under each cartoon in panel A. (D) Time course of *ndt80Δ* cells incubated in SM. Upper panels: differential interference contrast; lower panels: Zip1-GFP fluorescence in inverted contrast. Time points are the same as for panel B. The strong fluorescent signals in the 8- and 12-hour time points are polycomplexes.

## Visualization of meiotic nuclei *in situ* by cryo-ET

When this study started, we expected that the most conspicuous feature of pachytene cell cryo-tomograms would be SC densities. According to traditional-EM-based models, we expected to see 100-nm-wide zones of densely packed filaments, devoid of nucleosome-like particles. Our cryo-ET data did not have the density motifs expected of SCs, but instead revealed previously uncharacterized large filament bundles that assemble at the same time as the SCs. Herein we call the triple-helical units of these uncharacterized structures meiotic triple helices (MTHs) for the following reasons: first, these MTHs densities coincide with the appearance and disappearance times of Zip1-GFP fluorescence through meiosis; second, MTHs are absent in SM-treated haploid cells; third, MTH abundance greatly increases in pachytene-arrested cells. The term MTH does not imply that they are composed of three alpha helices (like a three-helix coiled coil). Also, the term does not imply knowledge of the structure's detailed subunit organization or helical parameters. MTH is a morphological term that is limited by the current data and the extent of analysis possible. This term may be revisited after the periodicity and twist along the longitudinal axis are better measured. We now present these experiments, which are summarized in Tables 1 and 2. Most of the cryo-EM imaging was performed by cryo-ET of cryosections; 2-D projection imaging was done when the cryosection attachment was non-ideal.

**Table 1. Pachytene yeast structural cell biology.**

| General cryo-ET observations | Result | Figure |
|---|---|---|
| Ladder-like densities | Not seen | various |
| Stacked-ladder densities in pachytene arrest | Not seen | various |
| Ordered loops of nucleosome-like particles | Not seen | various |
| 100-nm-wide nucleosome-free regions | At MTH bundles | various |
| **MTH general properties** | | |
| Presence after SM incubation | 4–6 hours | 2 |
| Presence in late meiosis | Absent/rare | S1A |
| Presence in haploid cells in SM | Absent | S1B |
| Assembly timing relative to Zip1-GFP | Coincident | various |
| Abundance during pachytene arrest | Increase | 3, S2 |
| Sensitivity to 1,6-hexanediol | Sensitive | 4D |
| Recovery after 1,6-hexanediol washout | Re-polymerizes | 4E |
| Subunit organization | Triple helical | 5 |
| Handedness | Right | 5D |
| Oligomerization state | Bundled | various |
| Packing density of MTH bundles | Crystalline | 6 |
| Thickness of bundles | $\geq 100$ nm | various |
| Presence in *spo11Δ* cells | Present | S3A |
| Presence in *zip1Δ* cells | Present | S3B |
| Presence in *red1Δ0* cells | Present | S3C |
| **F-actin perturbation during pachytene** | | |
| Rhodamine-phalloidin location | All cytoplasmic | S5A |
| Lifeact-mCherry foci location | All cytoplasmic | S5B, C |
| Treatment w/ 0.1% DMSO | MTH present | S6C |
| Treatment w/ 50 μM Lat-A + 0.1% DMSO | MTH absent | S6D |

To track the assembly of meiosis-related intranuclear complexes, we incubated WT cells in SM and then prepared self-pressurized-frozen samples every 2 hours for 8 hours. Self-pressurized freezing is a simpler and lower-cost alternative to conventional high-pressure freezing, which requires a dedicated machine that consumes large amounts of cryogen. In the self-pressurized freezing method, the sample is sealed in a metal tube and rapidly cooled in liquid ethane. The material in direct contact with the metal cools first and expands by forming

**Table 2. MTH detection by cryo-EM.**

| Cell line; treatment | MTH- positive | Cells imaged | |
|---|---|---|---|
| | | **cryo-ET** | **projection** |
| *spo11Δ/spo11Δ* | 9 | 20 | - |
| *zip1Δ/zip1Δ* | 14 | 53 | - |
| *red1Δ0/red1Δ0* | 12 | 20 | - |
| EW104 + DMSO | 15 | - | 24 |
| EW104 + Lat-A | 0 | 6 | 59 |
| EW104 + 5% 1,6-hexanediol | 21 | 43 | - |
| EW104 + 7% 1,6-hexanediol | 2 | 57 | - |
| EW104 + 7% 1,6-hexanediol, washout | 27 | 38 | - |
| WT in SM 8 hours | 2 | 36 | - |
| LY2 (W303) in SM 6 hours | 0 | 20 | 9 |

crystalline ice, which pressurizes the material in the center of the tube [22–24]. We then cut cryosections of these samples with nominal thicknesses of 70 or 100 nm. To increase the contrast, we collected most of our cryo-ET data using a Volta phase plate [25]. Cryotomograms of cell nuclei at the start of SM treatment (0 hours) did not contain any ~ 100-nm-or-larger structures expected of SCs (Fig 2A). These cell nuclei contained 10-nm-diameter nucleosome-like particles and ≥ 20-nm-wide multi-megadalton-sized complexes (megacomplexes, such as pre-ribosomes) like those seen in cryo-ET studies of yeast, insect, and HeLa nuclei [26–30]. After 2 hours in SM, a few cells had bundles of filaments (the MTHs). These MTHs each present a trefoil motif when viewed along their long axis (Fig 2B). At 4 and 6 hours in SM, the cell nuclei contained larger bundles of MTHs (Fig 2C and 2D). MTHs were rarely found in WT cells that were incubated for 8 hours in SM (S1A Fig).

We did not see any MTHs in more than 1,000 cryo-ET datasets of interphase and mitotic haploid *S. cerevisiae* cells that were incubated in rich media [26, 28, 31]. Recent studies showed that when yeast cells are stressed by glucose deprivation, components of the translation machinery form large cytoplasmic filaments [32, 33]. SM contains the less-preferred acetate as

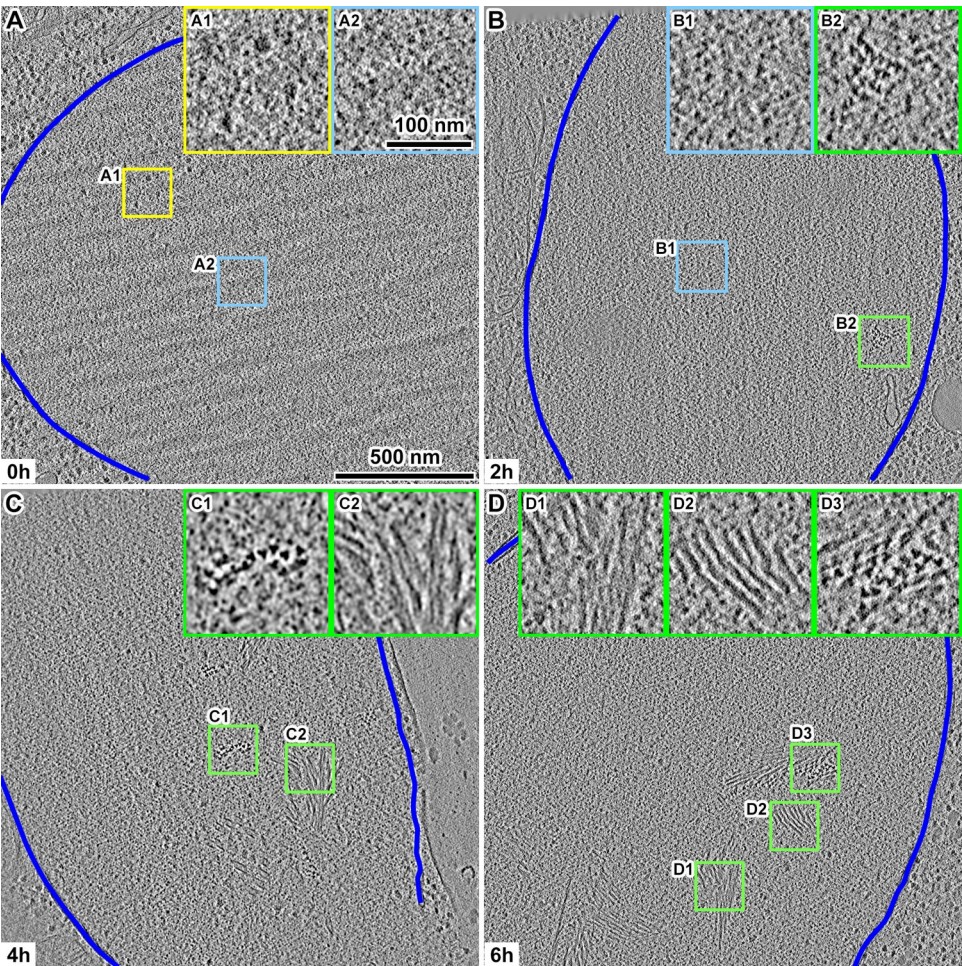

**Fig 2. Visualization of meiotic triple helix (MTH) bundles using cryo-ET.** Volta cryotomographic slices (10 nm; computational) of WT cell nuclei (A) before meiosis and after (B) 2 hours, (C) 4 hours, and (D) 6 hours in SM. The boxed areas are enlarged 3-fold in the corresponding insets. Inset A1 shows 3 megacomplexes. Insets A2 and B1 show nucleosome-like particles. Insets B2, C1, and D3 show bundles of MTHs presenting a trefoil motif. Insets C2, D1, and D2 show side views of bundles of MTHs. The nuclear envelopes are delineated with the dark blue lines.

the carbon source, which may induce stress-related structures that we had not seen before. To test if these intranuclear filament bundles are stress-related instead of being meiosis-related, we incubated the haploid strain LY2 in SM for 6 hours and then performed cryo-ET of their cryosections. Haploid cells treated this way did not contain any MTHs (S1B Fig), supporting the notion that these triple-helical structures are meiosis-related and their being called MTHs.

## Pachytene-arrested cells have more MTH bundles than wild type cells

The observation that MTH bundles and SCs appear and disappear at similar times suggests that these two structures are related. When incubated in SM, *ndt80Δ* cells arrest at pachytene with SCs, but continue to express SC proteins such that nearly every *ndt80Δ* cell has a poly-complex–a large aggregate of SC proteins that resembles side-by-side stacked SCs [9, 34–36]. Polycomplexes are rare in WT cells of the SK1 genetic background [17], so *ndt80Δ* cells offer a means to study the correlation between SC-related structures and MTH bundles. After an 8-hour incubation in SM, some *ndt80Δ* cells contained a bright fluorescent spot (polycomplex) and string-like signals (SCs) (Fig 3A). To verify that these cells contain polycomplexes, we prepared and then imaged plastic sections of negatively stained pachytene-arrested *ndt80Δ* cells. These cells had stain distributions expected of the polycomplex [9] (Fig 3B). Cryotomograms revealed that in pachytene-arrested *ndt80Δ* cells, MTH bundles are larger and more numerous (Fig 3C). These structures do not appear as stacked ladders (S2 Fig), but their dimensions (larger than 500 nm in some cases) are comparable with that of polycomplexes [20, 37].

## MTHs do not require SC assembly and are not composed of Zip1p or Red1p

SCs do not assemble unless the Spo11p endonuclease is present and active [38] and cannot exist without the central-region structural protein Zip1p [39] or the lateral-element structural protein Red1p [10]. To test if MTH assembly depends on the SC, we performed additional controls with null mutants that are unable to assemble SCs, namely, *spo11Δ/spo11Δ*, *zip1Δ/zip1Δ*, and *red1Δ0/red1Δ0*. Large MTH bundles were present in all three null mutants, just as in *ndt80Δ* cells (S3 Fig). We therefore conclude that MTH assembly does not depend on SC assembly and that MTHs do not contain the SC proteins Zip1p or Red1p.

## Ladder-like motifs and ordered chromatin are absent in meiotic yeast

We expected to see cryo-ET densities that resemble sections through a ladder-like structure, i.e., densely packed parallel filaments bridging a 100-nm-wide gap between two dense rails. However, we did not find ladder-like motifs in either pachytene-arrested *ndt80Δ* cells or in WT cells at any of the time points we sampled. We also did not find any of the side-by-side "stacked" ladder motifs expected of polycomplexes in pachytene-arrested cells. The ladder model is based on images of traditional EM samples. Furthermore, those traditional EM studies were done with projection images through tens of nanometers of cell mass, which may obfuscate the structural details of thick macromolecular complexes (S4A Fig). We tested this possibility by comparing projection images of cryosections with the increasingly thick cryotomographic slices from the same positions. In one example in which the MTHs were oriented obliquely to the cryosection surface, they appeared like filaments running parallel to the cryosection surface in the thickest tomographic slices (S4B and S4C Fig). Projections of this bundled MTH position have a chevron motif that only vaguely resembles a ladder.

Textbooks also depict SCs as having ordered chromatin loops, which are anchored at the SC lateral elements [40, 41]. Based on this model, we would expect to find numerous rows of nucleosome-like particles (10 nm wide), packed side-by-side and projecting from a linear

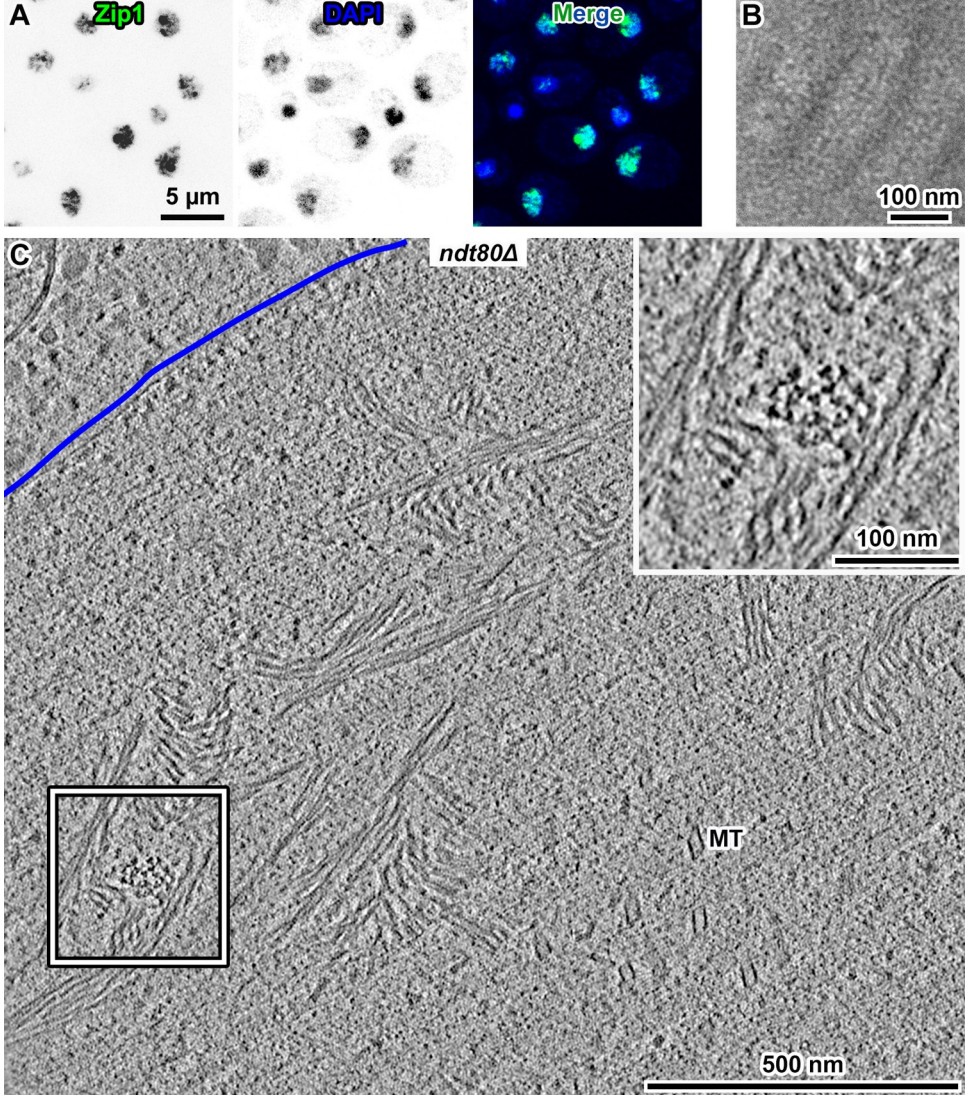

**Fig 3. Pachytene-arrested cells are highly enriched in MTHs.** (A) Fluorescence micrographs of pachytene-arrested *ndt80Δ* cells, after 8 hours in SM. Zip1-GFP fluorescence is found at both SCs (weak fluorescence) and polycomplexes (large, bright fluorescent bodies). The chromatin is stained by DAPI. (B) Plastic section showing a portion of a polycomplex in a pachytene-arrested *ndt80Δ* cell. (C) Volta cryotomographic 70-nm slice (computational) of the cryosection of a *ndt80Δ* cell at 8 hours in SM. The nuclear envelope (NE) is indicated by the blue line. The boxed area is enlarged 2-fold in the inset.

feature (the lateral element). We did not see periodic arrangements of nucleosome-like particles in any of the cells we imaged. This observation is consistent with the large body of data that chromatin has an irregular 3-D conformation *in situ* [42, 43].

## MTHs are not detected by actin-binding reagents, but depend on F-actin to assemble

A recent study also reported novel meiotic intranuclear filaments in yeast cells [44], though it is unclear if those filaments are triple-helical. Nevertheless, the similar abundance and bundling phenotypes suggest that the filaments observed by Takagi, *et al.* are the same triple-

helical structures we observed, so we will also refer to the meiotic filaments in their study as MTHs. In that study, immuno-EM showed that anti-actin antibodies [45] co-localized with the MTHs, which suggests that MTHs contain filamentous actin (F-actin). If true, then MTHs should be detectable by fluorescence-microscopy using F-actin probes because MTH bundles are thicker than and should have more signal than F-actin filaments and patches. We stained pachytene-arrested *ndt80Δ* cells with rhodamine-phalloidin and observed punctate signals in the cytoplasm (as expected), but not in the nucleus (S5A Fig). Another F-actin-visualization tool is the ABP140-GFP fusion protein, which previously revealed that F-actin localizes to the exterior of meiotic nuclei, but not inside [1, 46]. ABP140 is probably too large to diffuse through the nuclear pore to be an intranuclear-acting probe. Lifeact, the 17-amino-acid-long N-terminal peptide of the yeast ABP140 gene product, is also able to bind F-actin as a GFP fusion protein [47–49]. The Lifeact-GFP fusion protein is a 29-kDa fluorescent actin probe that is smaller than the size cutoff needed to passively diffuse through the nuclear pore complex [50]. We generated Lifeact-mCherry by integrating a mCherry tagging cassette at the 3' end of ABP140's 17th codon in the EW104 strain. This strategy allows the Lifeact-mCherry expression levels to be controlled by the natural ABP140 promoter [51]. Lifeact-mCherry did not form any structures in the nuclei of pachytene-arrested *ndt80Δ* cells (S5B and S5C Fig). Note that we cannot rule out that Lifeact-mCherry is excluded from the nucleus.

To further investigate the role of F-actin in meiotic nuclei, we treated cells with Latrunculin A (Lat-A), a drug that both inhibits actin polymerization and depolymerizes F-actin [52–54]. Fluorescence microscopy showed that Lat-A-treated meiotic *ndt80Δ* cells lost their cytoplasmic F-actin signals (S6A and S6B Fig). Cryo-EM of cryosectioned *ndt80Δ* cells revealed that MTHs were also absent after Lat-A treatment (0 out of 66 cryosections) but not after incubation with the DMSO carrier (15 out of 24 cryosections) (S6C and S6D Fig; Table 2). Therefore, MTH assembly either requires F-actin polymerization or is itself sensitive to Lat-A.

## MTHs and their bundles reversibly disassemble in 1,6-hexanediol

An earlier study showed that 1,6-hexanediol treatment can reversibly disperse yeast SC's central region proteins [55]. To test if MTHs have similar dispersal properties, we incubated WT cells with 1,6-hexanediol for 1 minute and then imaged them by fluorescence microscopy and cryo-ET. In the presence of 5% 1,6-hexanediol, the Zip1-GFP fluorescence signals appeared uniform in the nucleus while in 7% 1,6-hexanediol, the Zip1-GFP signal was largely cytoplasmic (Fig 4A). We then performed cryo-ET of WT cells that were treated with 0% (control), 5%, or 7% 1,6-hexanediol (Fig 4B–4D). MTH bundles were present in untreated cells and in 5% 1,6-hexandediol-treated cells (Fig 4B and 4C). We did not observe any MTHs in cells that were treated with 7% 1,6-hexanediol. Instead, some of the cells contained bundles of thinner filaments (Fig 4D). We attempted subtomogram 3-D classification of these 7%-1,6-hexanediol-resistant filaments but did not observe any meaningful class averages, either because these filaments are too conformationally heterogeneous or due to the limited contrast. The 7% 1,6-hexanediol-induced disassembly is reversible because MTH bundles reform rapidly after washout (Fig 4E).

## Subtomogram 3-D analysis of the MTHs

Cryotomographic slices of the MTHs in pachytene cells show two abundant 2-D motifs: densely packed 12-nm-wide filaments and trefoil-shaped densities that have a maximum width of 12-nm (Fig 5A and 5B). When we rendered sequential tomographic slices of MTHs that were perpendicular to the cryosection surface as a movie, they appeared to rotate, consistent with their helical nature (S1 Movie). To obtain a higher-resolution model of the MTHs,

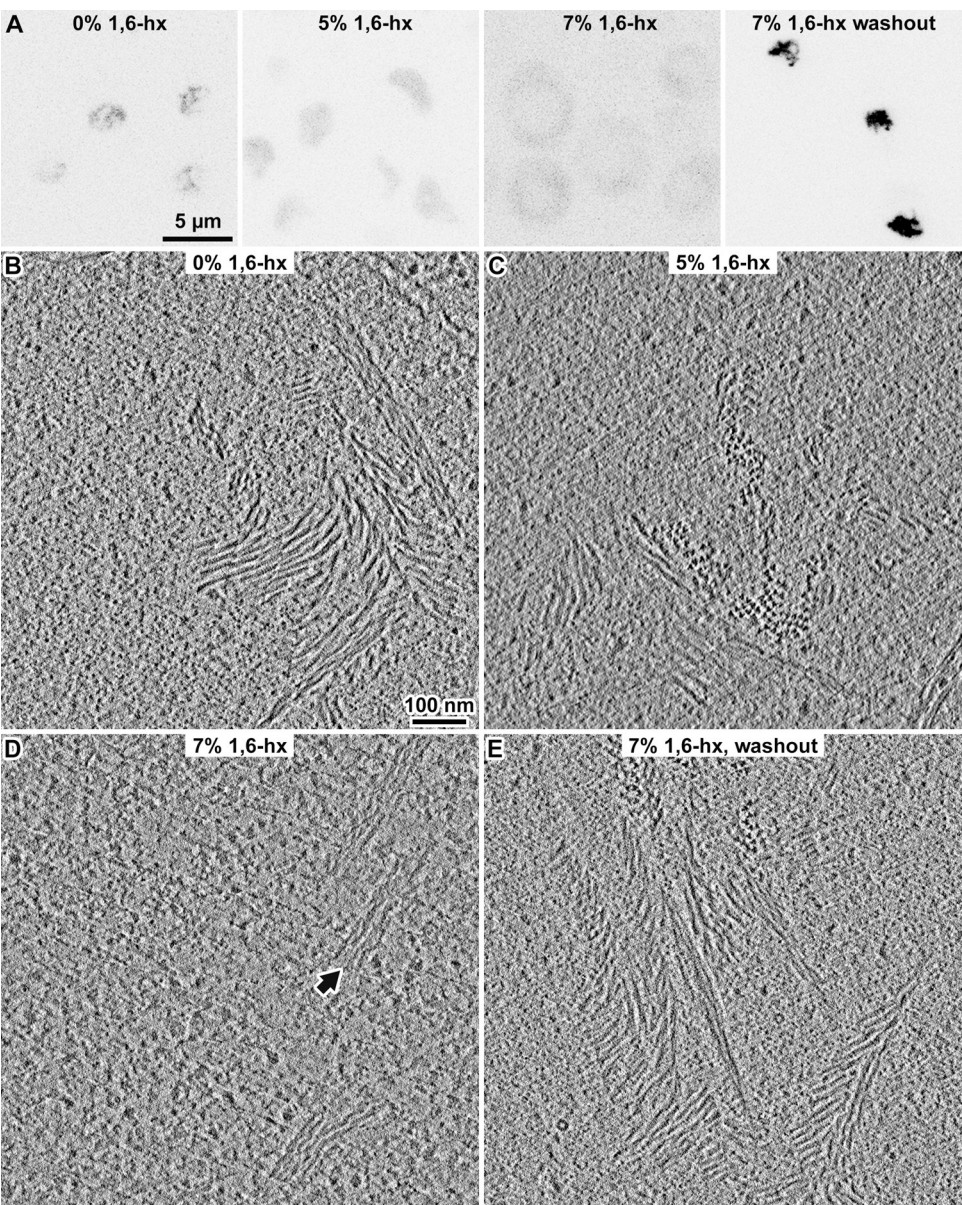

**Fig 4. The MTH and its bundles are sensitive to 1,6-hexanediol.** (A) Zip1-GFP fluorescence micrographs of WT pachytene cells treated with increasing concentrations of 1,6-hexanediol (1,6-hx), rendered with inverted contrast. The cells were harvested after 6 hours in SM and resuspended in SM with 0%, 5%, or 7% 1,6-hexanediol. The last subpanel shows cells treated with 7% 1,6-hexanediol, then incubated in fresh SM for 5 minutes. In the 7% 1,6-hx subpanel, the light-gray oval in the middle of the cell is the nucleus; most of the Zip1-GFP signal (darker pixels) is cytoplasmic. (B) Volta cryotomographic slice (10 nm; computational) of a WT cell in SM, showing MTH bundles. (C) Volta cryotomographic slice (10 nm; computational) of a WT cell in 5% 1,6-hexanediol. This cell contains a few MTH bundles. Approximately half the tomograms of such cells (21 of 43) contain these bundles. (D) Volta cryotomographic slice (10 nm; computational) of a WT cell in 7% 1,6-hexanediol. The arrow indicates a bundle of filaments of unknown structure. (E) Volta cryotomographic slice (10 nm; computational) of a WT cell after washout of 7% 1,6-hexanediol and then incubation in SM for 5 minutes.

we performed template matching, classification, and subtomogram averaging of MTH subvolumes from three strains studied here: DK428 treated with 5% 1,6-hexanediol after 4 hours in SM; NKY611 after 6 hours in SM; and EW104 after 8 hours in SM. We treated the MTH segments as independent particles, which is commonly done for single-particle analysis of helical

structures [56–58]. We template matched the MTH segments with a 30-nm long by 12-nm diameter cylinder reference. To lower the chances of missing some MTH segments, we used a low cross-correlation cutoff. We removed false positives by 2-D classification and then by 3-D classification (Fig 5C and 5D). We then 3-D refined the particles from the best 3-D classes to 33 Å resolution.

We could not distinguish between the MTHs from the three strains at the current resolution. Note that the polarity of the helix cannot be resolved in the present data, which may contribute to the averages' moderate resolution. The 3-D refined density maps confirm that the MTH is 12-nm thick (Fig 5D and 5E; S1 Movie), with a rise of ~ 5 nm. We estimate the pitch to be ~ 130 nm. Each strand within the MTH is ~ 5 nm thick. Our attempts to obtain averages of longer MTH segments using a taller mask produced averages that resemble featureless cylinders, suggesting that the MTHs have variable curvature, which prevents their alignment and averaging to higher resolution.

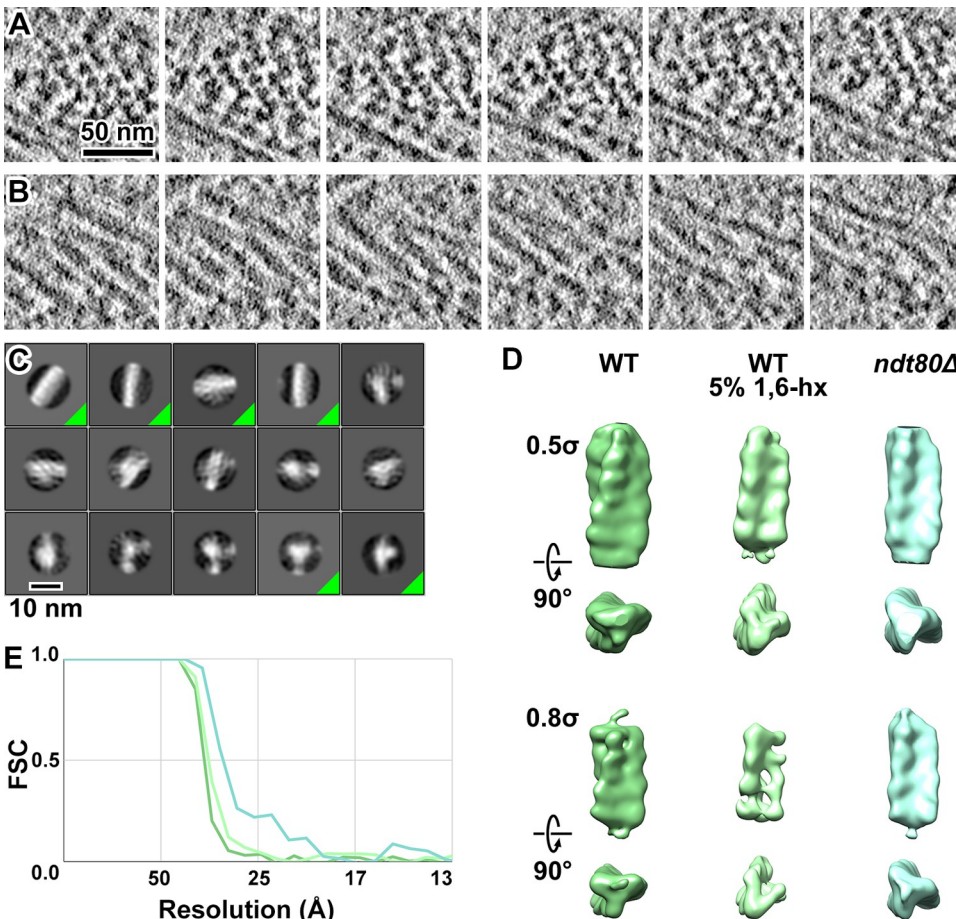

**Fig 5. MTHs are densely packed.** (A and B) Sequential Volta cryotomographic slices (12 nm; computational) through bundles of MTHs that present the trefoil and filamentous motifs. Both are from WT cells. (C) Class averages (2-D) of template-matched MTH segments, taken from subvolumes within *ndt80Δ* pachytene-arrested cells. The class averages corresponding to MTH that were used for subsequent analysis are indicated by a green triangle at the box's lower right. (D) Refined 3-D density map of MTH segments from WT, WT treated with 5% 1,6-hexanediol (1,6-hx), and *ndt80Δ* cells. They are rendered at two contour levels (0.5 σ and 0.8 σ) and viewed both perpendicular to (upper subpanels) and along (lower subpanels) the longitudinal axis. The handedness was confirmed by analysis of ribosomes; see S7 Fig. (E) Fourier-shell correlation (FSC) plots of the MTH segments from three conditions, color-coded similarly to panel B. The resolutions of the averages are ~ 33 Å, based on the "Gold standard" FSC = 0.5 criterion.

All three subtomogram averages revealed right-handed MTHs. The handedness of electron tomograms is ambiguous unless a known chiral structure is imaged with the same conditions as the structure of interest [59]. Ribosomes are abundant in cellular cryotomograms and have asymmetric features that are visible in *in situ* subtomogram averages [27, 60]. We therefore performed template matching, classification, and subtomogram averaging of cytoplasmic ribosomes from *ndt80Δ* cell cryotomograms, which had the highest contrast. One 80S ribosome class average shows that the characteristic "beak" motif is oriented in the direction expected of the ribosome (S7 Fig). This experiment confirms that the MTH is right-handed.

## MTH bundles are ordered and helical

To better understand how MTHs pack together, we attempted to make 3-D models of MTH bundles. Because the remapping of subtomogram averages can reveal higher-order motifs in chromatin structure [29, 61], we first remapped the subtomogram averages of the MTH segments back into the tomogram. This approach failed because template matching and classification are imperfect–they generate false negatives, which causes the filaments to appear discontinuous. As an alternative, we manually modeled each MTH as rod (Fig 6A and 6B). The 3-D models show that MTHs form domains of largely parallel filaments that appear ordered. To better characterize this order, we generated power spectra by Fourier transforming the tomographic slices of the largest WT filament bundles (Fig 6C and 6D). These power spectra revealed broad peaks corresponding to spacings between 14 and 22 nm, consistent with hand-measured center-to-center distances between neighboring MTHs. This tight packing is consistent with the absence of macromolecular complexes, such as nucleosome-like particles, in between the MTHs. The peaks appear elongated instead of circular, meaning that MTHs pack together with helical order, namely, an MTH bundle is a helix of helices. To

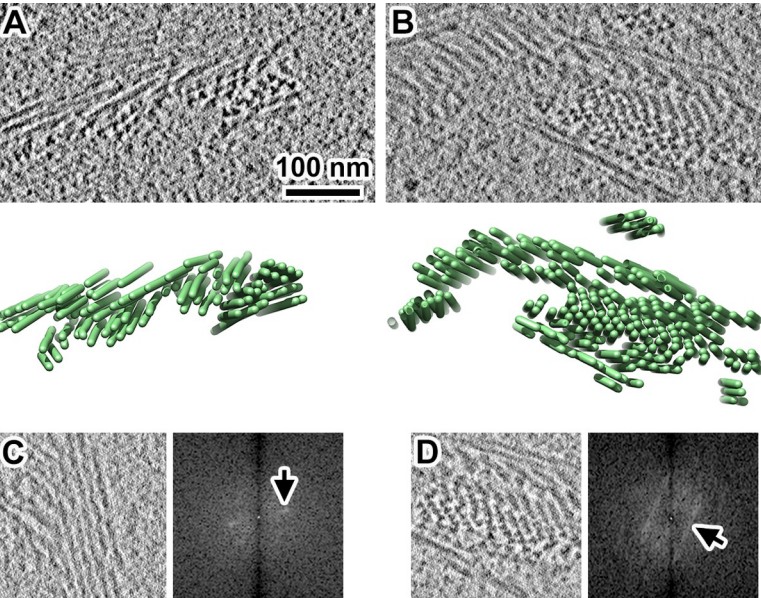

**Fig 6. Yeast MTHs form ordered bundles.** (A and B) Upper: Volta cryotomographic slices (10 nm; computational) of WT cells that were frozen after a 4-hour incubation in SM. Lower: 3-D models of the distribution of MTH bundles. Each MTH is rendered as a tube. (C and D) Volta cryotomographic slices (left, 10 nm; computational) and power spectra (right), taken at regions with many MTHs. Panel D shows the same MTH bundle as panel B, but from a different position along the Z axis. Arrows point to elongated peaks in the power spectra, which correspond to approximately (C) 14 nm and (D) 22 nm spacings.

characterize the distribution of the MTHs, we attempted to reconstruct as much of a single cell as possible by serial Volta cryo-ET of cryosections [62]. We successfully reconstructed six sequential sections from one ndt80Δ cell (S8 Fig), which represents approximately one third of a nucleus (assuming a spherical shape). These reconstructions show that all the MTHs in a 420-nm-thick sample of the cell were associated in ordered bundles. Indeed, we have not yet found an example of an isolated MTH in our other tomograms.

## Discussion

### Meiotic yeast nuclei appear very different in the lens of cryo-ET

In textbooks, the two most prominent features of pachytene nuclei are the ladder-like SC and the ordered chromatin loops that project from the SC's lateral elements (Fig 7A). The ladder model comes from traditional EM studies, starting with the discovery of the SC sixty-five years ago [11, 12]. This model has been supported by imaging studies of multiple organisms, including worms [63], beetles [64], flies [65], and plants [66]. When yeast is subjected to traditional EM sample preparation [67, 68], ladder-like densities are rarely seen. In contrast, if the cell wall is enzymatically removed prior to traditional EM heavy-metal staining [9, 14], then ladder-like motifs are abundant in the subsequent images, consistent with the model of the SC from other organisms. Based on these models, one would expect our cryotomograms of

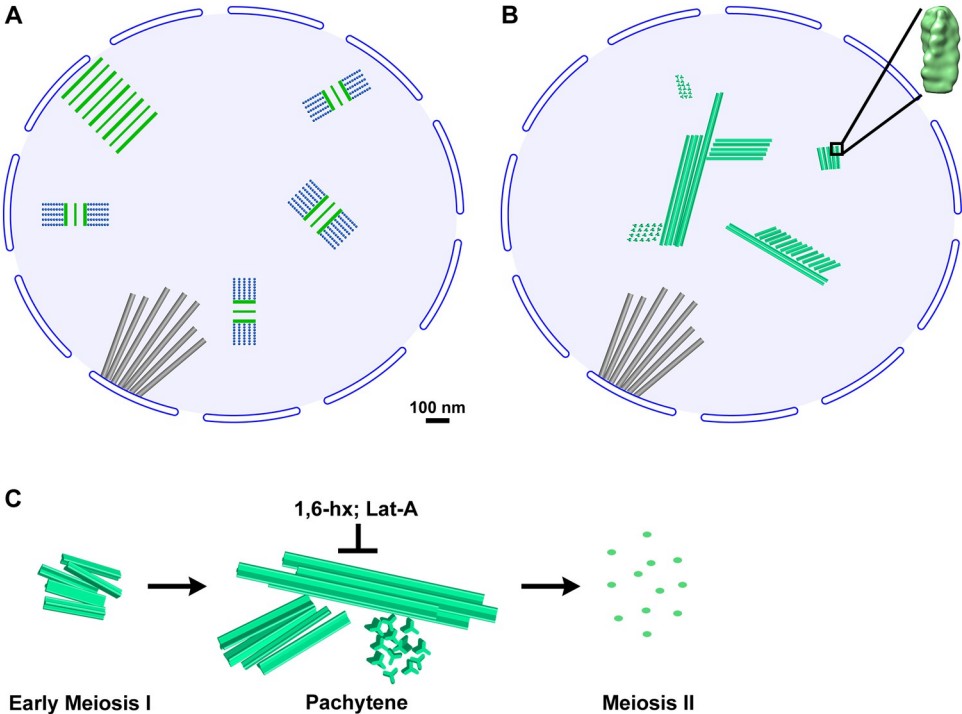

**Fig 7. A cryo-ET model of the meiotic yeast nucleus.** (A) Traditional EM model of a section through a pachytene yeast nucleus. The largest structures are the nuclear-microtubule array (gray), SC sections (green and blue), and the polycomplex (parallel green lines). In the traditional EM model, the SC and polycomplex are densely packed structures. The nucleosomes (blue dots) project as ordered arrays from the SC. All structures are illustrated at the approximately correct scale. (B) In the cryo-ET model based on the cryo-ET analysis here, the bundles of MTHs (green polymers) are the only large, ordered structures in pachytene-arrested cells. The MTH bundles coexist with spindle microtubules (gray tubes) and SCs, whose cryo-ET structure remains unknown. The chromatin, which is irregular, is represented by the uniform light-blue background. (C) MTHs form large, densely packed bundles in pachytene. These bundles are sensitive to 1,6-hexanediol (1,6-hx) and Latrunculin A (Lat-A). The MTHs disassemble after pachytene.

pachytene yeast to contain two types of ordered motifs: 100-nm-wide ladder-like densities and densely packed rows of nucleosome chains. We did not see any densities that resemble these two ordered motifs, even when considering that previous EM studies were largely done by projections. Note, we are not arguing against the existence of SCs or that SCs do not have a ladder-like organization. Our data do not imply–and have not been interpreted–as evidence of the absence of SCs. The SCs are present in the cells, based on the presence of Zip1-GFP signals. Our study shows that when pachytene yeast are imaged by cryo-ET, the structural motifs expected of SCs are not visible. Given the advantages of cryo-ET over traditional EM, our study does provide important constraints for the organization of SCs in unstained, unfixed yeast. We did not observe any structures that resemble what has been depicted of SCs in the literature. Namely, we did not observe any structure that appears packed to crystalline density, except for the MTH bundles (see below). Note that we are not implying that the SCs are built from MTHs because these are clearly distinct structures as mutants that cannot assemble SCs are still able to make MTHs. Also note that there is no consensus on whether the SC is crystalline or not, as seen in papers that depict SCs with [55, 65] and without [69, 70] order.

## MTH bundles are the largest ordered structures in pachytene cells

We observed large, ordered bundles of MTHs, which are absent in textbook models of meiotic cells. Our cryo-ET data suggest a model of the pachytene yeast nucleus in which intranuclear microtubules and MTH bundles are the most-densely packed structures and coexist with the SC, whose cryo-ET structure remains unknown (Fig 7B). The MTH bundles generate weak layer lines in Fourier transforms, which suggests that they pack into a helical array such as a cholesteric liquid crystal [71]. To our knowledge, no previous meiosis studies have reported MTHs in the nucleus or cytoplasm. The MTH is approximately 12-nm thick at its widest point and has a 5-nm rise and a 130-nm pitch. Each of the three strands is approximately 5-nm thick. These strands are approximately five times thicker than alpha helices (~ 1 nm) so the MTH is not a three-helix coiled coil. If the approximately 5-nm spacing along the long axis arises from 5-nm-wide globular proteins, then the subunit can be accounted for by many types of proteins, including actin, tubulin, and dimers of smaller proteins. Higher-resolution analysis is needed to define the subunit structure. The center-to-center separation between MTHs varies from 14 nm to at least 22 nm. In unperturbed cells, we did not detect isolated MTHs or thinner subassemblies, meaning that assemblies smaller than MTH bundles are rare. The rarity of smaller MTH assemblies and subassemblies suggests that the intra- and inter-MTH interactions are stable.

## MTH bundles have SC-like properties but lack key SC proteins

Our study raises a major question: what is the composition of the MTH? Their identification will enable studies that address MTH's function, assembly and disassembly, relationship to actin polymerization, and the possible existence in other organisms. The MTH's SC-like properties may help their future identification and characterization. MTH and SC components share similar assembly-disassembly timings and abundance in pachytene cells (Fig 7C). They are both sensitive to 1,6-hexanediol and are localized in the nucleus, though we did see occasional MTHs in the cytoplasm. They both form large intranuclear bodies that assemble in meiosis I, though only MTHs are visible by cryo-ET. These similarities led us to initially hypothesize that the MTHs are part of the SC. This hypothesis was falsified by our observation of MTH bundles in mutant cells that cannot assemble SCs (*spo11Δ*) or that cannot produce one of the structural proteins (*zip1Δ* and *red1Δ0*). MTH bundles are therefore not part of the SC and their assembly does not depend on SCs.

Using two popular F-actin probes, rhodamine-phalloidin and Lifeact, our experiments did not detect intranuclear actin. We did find that the MTHs are sensitive to the actin-depolymerizing drug Lat-A. In contrast, Takagi *et al.* provided immuno-EM evidence that the similar filaments they saw (probably MTHs) contain actin [44]. While MTHs cannot be F-actin, which would appear double helical, they could instead be a non-canonical triple-helical polymer of actin that is sensitive to Lat-A and is bound by anti-actin antibodies in immuno-EM experiments, but unable to bind rhodamine-phalloidin and Lifeact. We are unaware of any precedent for triple helical actin polymers. A recent cryo-ET study of bundled F-actin *in situ* revealed an average nearest-neighbor spacing of ~ 12.3 nm [72] with filaments as close as 8 nm. While some of the MTHs are packed to ~ 12 nm apart, none were seen 8 nm, possible because MTHs are thicker than F-actin.

Note it is possible that the MTHs may not be directly involved in meiosis but are instead a protein that is at a sufficient concentration or has the right biochemical modifications to form helices in pachytene because it is known that many proteins can form a helix under the right conditions [73–75]. These MTHs also have lateral interactions that allow them to pack with crystalline density. Their sensitivity to 1,6-hexanediol suggests that the polymerization both within and between MTHs are based on hydrophobic interactions. Further work will be needed to determine the identity of the MTH's subunits and their potential function.

## Materials and methods

### Strains

Strains DK428 (WT) and EW104 (*ndt80Δ*) were gifts from the Kaback lab. NKY2292, NKY2293, NKY2296 (*ndt80Δ/ndt80Δ*), NKY2460 (*zip1Δ/zip1Δ*), and NKY2535 (*spo11Δ/spo11Δ, ndt80Δ/ndt80Δ*) were gifts from the Kleckner lab. The haploid LY2 strain was a gift from the Lacefield lab. The strains' key genotypic features and experimental roles are summarized in S1 Table.

### Construction of Red1 deletion and Lifeact-mCherry strains

The new strains' genotypes are also summarized in S1 Table. Sequence analysis and primer design was done in Benchling [76]. All the plasmids were from Addgene (Watertown, MA) and primers (S2 Table) were from IDT (Coralville, IA). Sequencing was done by Bio Basic (Bio Basic Inc., Singapore). Red1 deletion mutants (*red1Δ0::KANMX* and *red1Δ0::URA3*) of *S. cerevisiae* haploid *ndt80Δ* strains NKY2292 and NKY2293 were created by PCR-mediated gene replacement with the *KANMX* and *URA3* markers, respectively [77]. Plasmids pFA6a-link-yoTagRFP-T-Kan (Addgene 44906) and pFA6a-link-yomCherry-CaURA3 (Addgene 44876) were purified with QIAprep spin miniprep kit (QIAGEN, Hilden, Germany), digested with restriction enzyme *Sac*I-HF (NEB, Ipswich, MA), and served as templates for the *KANMX* and *URA3* deletion cassettes, respectively. The extension PCR amplicons have 40 bp of homologous sequences upstream and downstream of *RED1* and were amplified with Q5 polymerase (NEB). Haploid cells were transformed by the lithium acetate/single-stranded carrier DNA/PEG4000 method [78]. MATa cells (NKY2292 background, *red1Δ0::KANMX*) cells were recovered in 1 mL YPD (1% yeast extract, 2% Bacto peptone, 2% D-glucose) at 30˚C in a ThermoMixer C (Eppendorf, Hamburg, Germany) for 3 hours (200 RPM) before plating. MATα cells (NKY2293 background, *red1Δ0::URA3*) were selected in synthetic defined −uracil media. DNA was extracted from transformants via the LiAc-SDS gDNA method [79]. Single MATa and MATα *red1Δ0* colonies grown on YPD plates were mixed and spread onto double-selection plates (+G418, synthetic defined medium, −uracil, with monosodium glutamate as the nitrogen source). For Lifeact-mCherry strain construction, pFA6a-link-yomCherry-Kan

(Addgene 44903) were linearized with *Sac*I-HF (NEB). PCR tagging cassettes were amplified from this plasmid to contain up to around 50 base pairs of homology sequences with upstream and downstream of the 17[th] codon of ABP140. These PCR cassettes insert in-frame at the 3' end of the 17[th] codon of ABP140. The PCR cassette was then transformed into EW104 to create LGY0069 using the lithium acetate method [80]. All strains were also validated by PCR and Sanger sequencing.

## Cell culture and synchronization

Most experiments were done at room temperature (23˚C) unless noted otherwise. Cells were sporulated using a previous protocol [17, 20], modified as follows. A single colony from a YPD plate was inoculated into 3 mL YPD medium and grown overnight at 30˚C with shaking at 220 RPM in a 15 mL culture tube. Cells were pelleted at $3,000 \times g$ for 5 minutes, washed once with ddH$_2$O (deionized and distilled water, $\geq$ 18 M$\Omega$·cm) and then resuspended in 20 mL YPA (0.5% yeast extract, 1% Bacto peptone, 1% potassium acetate; pre-sporulation medium) in a 250 mL flask, which was used for all subsequent incubations. After a 3-hour pre-sporulation incubation at 30˚C, the cells were pelleted, washed three times with ddH$_2$O, and then resuspended in 20 mL sporulation medium (SM– 2% potassium acetate) with 0.5 mL amino-acid supplement (0.04 g/L L-proline, 0.02 g/L L-lysine, 0.08 g/L L-tyrosine, 0.2 g/L L-histidine, 0.2 g/L L-leucine, 0.2 g/L L-methionine, 0.2 g/L L-tryptophan, 0.2 g/L L-arginine, 0.4 g/L Adenine, 0.4 g/L Uracil, 1 g/L L-phenylalanine; Sigma LAA21; Sigma, St. Louis, MO). Sporulation was done at 30˚C with shaking.

For time course experiments: A single colony from YPD plate was inoculated into two 12 mL aliquots of YPD medium and grown overnight at 30˚C with shaking at 220 RPM in 50 mL conical tubes. Cells were pelleted at $3,000 \times g$ for 5 minutes, washed once with 0.5 volumes of ddH2O and resuspended to 1 mL YPA, and transferred into a 2 L flask containing 200 mL YPA. After 3 hours of pre-incubation at 30˚C, the cells were transferred to four 50 mL conical tubes, pelleted at $3,000 \times g$ for 5 minutes, washed twice with 0.5 volumes of ddH$_2$O, and resuspended in 0.25 volumes of SM. The resuspensions were then added to a 2 L flask containing 120 mL SM with 4 mL amino-acid supplement. Sporulation was done at 30˚C with shaking. At each time-point, 20 mL culture was withdrawn and pelleted at $3,000 \times g$ for 5 minutes. The cells were then resuspended in 700 µL PBS and fixed with 4% paraformaldehyde (Electron Microscopy Sciences, 15714) for 1 hour at 30˚C with shaking at 220 RPM.

## Treatment with 1,6-hexanediol

Six molar 1,6-hexanediol (Sigma 88571) was diluted with ddH$_2$O to a final concentration of 10% or 14% (w/v). DK428 cells (1 mL) were incubated for 6 hours in SM and then transferred to a 1.5 mL microfuge tube and centrifuged at $3,000 \times g$ for 5 minutes. The cell pellet was resuspended with 0.5 mL ddH$_2$O. For 5% or 7% (0.42 M or 0.59 M, respectively) hexanediol treatment, 0.5 mL of 10% or 14% hexanediol solution was added to 0.5 mL of cell suspension and then mixed by pipetting for 1 minute. Fluorescence microscopy was performed on 4 µl of 1,6-hexanediol-treated cell mixture that was spread onto a microscope slide.

## Treatment with Latrunculin A

EW104 (*ndt80Δ*) cells were sporulated for 6 hours and then treated with 0.5% DMSO (Sigma D2650; the carrier) or 50 µM Latrunculin A (Abcam 144290; Abcam, Cambridge, UK) for 2 hours, at 30˚C with shaking at 220 RPM. For cryo-ET, the cells were pelleted by centrifugation at $3,000 \times g$ for 5 minutes at 23˚C, then subjected to self-pressurized freezing and vitreous sectioning. For LM, the cells were fixed with 3.7% formaldehyde (Sigma F8775) in SM for 1 hour

at 30˚C, then washed with ddH$_2$O three times. The fixed cells were resuspended in 500 μl PBS + 0.1% Triton-X100 with 1 μl rhodamine phalloidin reagent (Abcam 235138) and incubated at 23˚C with rotation for 20 minutes. The cells were washed with ddH$_2$O and imaged by fluorescence microscopy.

### Fluorescence microscopy of time course experiments

Sporulating DK428 and EW104 cells were collected hourly until for 8 hours, and then at 12 hours after transfer to SM; see above for details. Two μL of concentrated cells were mounted onto a glass microscope slide. A coverslip was placed above the sample and sealed with nail polish. Images were recorded at 23˚C using a Zeiss LSM900 inverted microscope with Objective Plan-Apochromat 63x/1.4 Oil objective lens (Zeiss, Jena, Germany). The cells were illuminated with a 488 nm laser. A subset of cells was stained with Vectashield-DAPI (Vector Laboratories, Inc. #H-1200-10, Burlingame, CA) and then imaged using a PerkinElmer Ultraview Vox (Waltham, MA) spinning-disc confocal microscope with a 100× oil-immersion objective lens.

### Fluorescence microscopy of Lifeact-mCherry expressing cells

Pachytene-arrested LGY0069 cells were concentrated and 1.5 μL of cells were mounted onto an SM agarose pad to enable live-cell imaging. A coverslip was placed above the sample and sealed with Vaseline. Images were recorded at 30˚C using an Olympus IX 83 inverted microscope with an Olympus UPLSAPO60xO objective lens (Olympus, Tokyo, Japan). Temperature control was monitored using Oko lab stage insert chamber (Olympus) to ensure the set-up was kept at 30˚C. Laser diodes (488 nm and 561 nm) were used to drive fluorescence signals from Zip1-GFP and Lifeact-mCherry, respectively.

### Plastic sectioning

EW104 (*ndt80Δ*) cells were harvested after an 8-hour SM incubation and then fixed with 4% paraformaldehyde in SM for 1 hour at 30˚C. The fixed cells were washed three times with water and then partially lysed with a yeast nuclei isolation kit (Abcam 206997) using the following modified protocol, using buffers from the kit unless otherwise stated. The cell pellet was resuspended in 1 mL Buffer A (with 10 mM dithiothreitol added), then incubated for 10 minutes in a 30˚C water bath. Cells were pelleted at $1,500 \times g$ for 5 minutes at 23˚C and resuspended in 1 mL Buffer B. After addition of 10 μl lysis enzyme cocktail, the cells were incubated at 30˚C for 15 minutes with shaking at 220 RPM and then pelleted at $1,500 \times g$ at 4˚C. The cell pellet was resuspended in Buffer C, incubated at 23˚C for 10 minutes, and then washed in ddH$_2$O. Plastic sections were prepared from this cell pellet using a published protocol [35] that was modified as follows. The partially lysed cells were dehydrated at 23˚C with sequential treatments: 70% ethanol (20 minutes) → 90% ethanol (20 minutes) → 95% ethanol (20 minutes) → 100% ethanol (20 minutes, twice) → 50% ethanol: 50% LR White (hard grade, London Resin Company) (1 hour) → 100% LR White (overnight). The cells were transferred into a gelatin capsule filled with LR White resin and then polymerized at 50˚C for 24 hours. Plastic sections (80 nm nominal) were cut in a Leica Ultracut UCT/FCS (Leica Microsystems, Vienna, Austria) with a glass knife, floated onto a water trough, collected with a 200 mesh continuous-carbon EM grid, and stained with UranyLess (EMS 22409; Electron Microscopy Sciences, Hatfield, PA) at 23˚C for 1 hour. Projection images of plastic sections were recorded with a Tecnai T12 transmission electron cryomicroscope (TFS) using a 4k × 4k Gatan UltraScan CCD camera (Gatan, Inc., Pleasanton, CA).

## Self-pressurized freezing and vitreous sectioning

Cells were self-pressurized frozen using a method modified from [23]. Sporulating cells were pelleted by centrifugation at $3,000 \times g$ for 5 minutes at 23˚C. The cell pellet was mixed with 60% dextran (Mr = 40 kDa, Sigma 31389) in SM. The cell/dextran mixture was quick-spun for 30 seconds to pop the bubbles and then loaded into a copper tube (0.3 mm inner diameter) with a syringe-type filler device (part 733–1, Engineering Office M. Wohlwend GmbH). Both ends of the copper tube were tightly clamped with flat-nosed pliers and then dropped into liquid ethane. Before loading into the microtome, the tubes' clamped ends were cut off with a tube-cut device (Part. 732, Engineering Office M. Wohlwend GmbH) operated in liquid nitrogen.

EM grids were pre-coated with 4 μl of 0.1 mg/ml BSA with 10-nm colloidal gold beads (Sigma G1527) in ddH$_2$O and air dried for either 3 hours or overnight before use. Vitreous sections of 70 or 100 nm nominal thickness were cut with a diamond knife (16DIA.DCO3530; Diatome, Nidau, Switzerland) in a Leica UC7/FC7 cryo ultramicrotome operated at −150˚C and augmented by dual micromanipulators [62, 81, 82]. The ribbon was attached to the gold pre-coated EM grid by charging with a Crion device (Leica Microsystems) in charge mode for 1 minute. The grid was stored in liquid nitrogen until imaging.

## Cryo-ET imaging and reconstruction

Tilt series were collected on Titan Krios (TFS) transmission electron cryomicroscopes equipped with a Volta phase plate, using either a Falcon II direct-detection camera (TFS) or a BioQuantum K3-GIF camera (Gatan, Inc.). Falcon II images were recorded in integration mode with Tomo4 (TFS). K3 dose-fractionated super-resolution movies were recorded under the control of SerialEM [83], with summed frames saved every 100 milliseconds. Additional imaging parameters are listed in S3 Table. Cryotomogram reconstruction was performed semi-automatically using the batch processing function of the IMOD software package [84–86]. The tilt series were aligned using either the gold beads as fiducials or by patch tracking if the field of view did not have enough gold beads. Contrast transfer function compensation (phase flipping, using IMOD *ctfphaseflip*) was done for the defocus phase-contrast data but not for Volta phase-contrast data. All cryotomograms analyzed in this paper are detailed in S4 Table.

## Cryo-EM projection imaging of cryosections

For projections, 9 images were recorded of haploid-cell cryosections with Volta phase contrast on a Falcon II camera. 83 projections were recorded with defocus phase contrast on a K3-GIF camera. See the bottom of S4 Table for additional details.

## Template matching and initial model

References were created with the Bsoft program *beditimg* [87] while masks were created with *beditimg* and the RELION (REgularised LIkelihood OptimisatioN) program *relion_mask_create* [88]. Uniformly spaced search points were seeded with the PEET program *gridInit* (Particle Estimation for Electron Tomography) [89, 90]. Template matching was done using PEET. We used different template-matching strategies for MTHs and ribosomes as detailed in S5 Table. MTH filaments are long and curvy, so they had to be treated as short helical segments. The grid points fit in a rectangular box that enclosed the MTH bundles. The search was done with 10˚ angular increments and using a Gaussian low-pass filter that attenuates spatial frequencies beyond 10-nm resolution. To minimize the influence of nearby particles, the subtomograms

were masked by a cylinder (MTH) or sphere (ribosome) that was extended 1 to 2 nm beyond the reference's outer-most pixels. Only one round of template matching was done, i.e., the reference was not refined. To lower the number of false negatives, we used a very low cross-correlation cutoff of ~ 0.1 to 0.3. This criterion resulted in more than 10-fold excess of false positives, as assessed by comparing the number of hits that remained after the classification process described below. A second round of template matching was done using the knowledge gained from subtomogram classification analysis as detailed in the next section.

## Subtomogram classification

Subtomogram classification and averaging were performed using RELION's subtomogram-analysis routines [91–93]. Two-dimensional classification of the projected subtomograms revealed class averages that clearly distinguished between the MTH densities and globular structures [91]. For 3-D classification, a 24-nm tall, 7-nm diameter cylindrical rod was used as a reference. To suppress the influence of neighboring particles, the densities were masked with a 16-nm tall, 19-nm-diameter soft-edged cylinder. No symmetry was enforced. Some of the resultant 3-D class averages were clearly MTHs, based on their dimensions and helical appearance. Other 3-D class averages had poorly defined shapes and were excluded as false positives. To improve the detection of false positives, the dimensions (length and radius) of the mask were varied. A taller cylindrical mask (24 nm) enabled the detection of rod-shaped densities that are shorter than the MTH. This new mask also probably caused the removal of the template-matching hits of MTHs close to the cryosection surface, where cryosectioning artifacts and contaminants generate spurious densities. Two rounds of 3-D classification using this taller mask pruned the dataset to a few thousand subtomograms. These subtomograms were then subjected to "gold-standard" 3-D autorefinement [94], but using a shorter, 16-nm-tall mask. Use of the shorter mask resulted in class averages that have higher-resolution features, such as 5-nm periodic density bumps along the helical axis. These features were absent when taller masks were used, probably because the MTHs have variable curvatures, which make the MTH segments more heterogeneous. The resultant higher signal-to-noise-ratio class average allowed us to measure the diameters and design a skinnier cylindrical template more accurately. Furthermore, this shorter class average (and correspondingly shorter cylindrical mask) was used for a second round of template matching, but with a denser search grid to maximize the number of MTH-containing subtomograms detected. The second set of template-matching hits was filtered at a cross-correlation coefficient cutoff of 0.2. This cutoff was determined subjectively by incremental increases, followed by inspection of the hit positions in the cryotomograms. The cutoff was chosen such that a nearly equal number of obvious nucleoplasmic false positives, such as nucleosome-like particles, were also included. These hits were 2-D and 3-D classified again.

Three-dimensional refinement was done using 841, 1,513, and 4,456 MTH segments from NKY611, DK428, and EW104 cells, respectively. Preliminary 3-D refinement on EW104 MTH segments produced a Fourier shell correlation curve that did not drop to zero in the highest-resolution range. This artifact was caused by new "duplicate" particles that were generated by the 3-D classification translational search [29]. These duplicates were removed by re-running template matching with PEET, using only the coordinates of the 3-D classified particles. The remaining 4,161 MTH segments were re-refined in RELION. FSCs were plotted with Google Sheets (Alphabet, Inc., Mountain View, CA). Subtomogram average visualization was done with UCSF Chimera [95].

## Fourier analysis

To maximize the signal-to-noise ratio for Fourier analysis of MTHs, positions with the largest bundles of MTHs were selected. Tomographic slices (10 nm) were exported as TIFF files using the IMOD slicer tool and then read into FIJI v2.0.0 [96]. A ~512-pixel-wide rectangular selection was made around the bundled MTHs and then Fourier transformed using the FFT tool. The distance d*, in inverse pixels, between a peak's center of mass and the origin was measured using the straight-line tool. This distance was converted to real space units with the equation

$$d = 512 \times pixel/d*$$

In this equation, "pixel" is the real-space pixel size (0.58 nm).

## Ribosome density simulation

The ribosome density map was simulated from the 80S ribosome crystal structure [97] using the Bsoft program *bgex* [87]. An image of the mirrored map was created directly in UCSF Chimera (Pettersen, 2004} with the command *vop zflip*.

## Supporting information

**S1 Fig. MTHs are not detected in late meiosis or starved haploid cells.** (A) Volta cryotomographic slice (12 nm; computational) of a diploid WT yeast cell after an 8-hour incubation in SM. PSM, prospore membrane; NE, nuclear envelope. (B) Volta cryotomographic slice (12 nm; computational) of a haploid cell (strain LY2, W303 background) incubated 6 hours in SM. The insets show 3-fold enlargements of the nucleoplasm boxed in each panel.
(TIF)

**S2 Fig. Examples of MTH bundles in pachytene cell nuclei.** Volta cryotomographic slices (10 nm; computational) of 12 examples of *ndt80Δ cell* nuclei after 8 hours in SM. The arrowheads in the lower-left panel indicate nuclear microtubules.
(TIF)

**S3 Fig. The MTH is present in null mutants of Spo11, Zip1, or Red1.** (A) Volta cryotomographic slice (6 nm; computational) of a *spo11Δ* cell in SM, showing intra-nuclear MTH bundles. (B) Volta cryotomographic slice (12 nm; computational) of a *zip1Δ* cell in SM, showing MTH bundles inside the nucleus. (C) Defocus phase-contrast cryotomographic slice (12 nm; computational) of a *red1Δ0* cell in SM, showing intra-nuclear MTH bundles. Insets show two-fold enlargements of the boxed areas. The contrast in panel C appears different from panels A +B because of the different contrast mechanism (defocus phase contrast instead of Volta phase contrast) and because the data was recorded on an electron-counting camera and by zero-loss energy filtering. This difference in contrast is also evident in (D) the *ndt80Δ* control cell cryotomogram, imaged in similar conditions as for panel C.
(TIF)

**S4 Fig. Comparison between tomographic slices (computational) and projections.** (A) Cartoon of comparison between projection and tomographic slices (computational) of various thicknesses. Thinner tomographic slices reveal details like microtubule protofilaments. Thicker tomographic slices resemble projection images. (B) In some projections, contiguous MTH structures appear to orient approximately perpendicular (left-right) to the orientation of the filaments (out of the plane). (C) An MTH bundle position in a cell that has a chevron-like motif in projection. The thinner tomographic slices (computational) show that the underlying

structure is a set of densely packed MTHs that are oblique relative to the Z axis.
(TIF)

**S5 Fig. F-actin does not localize to nuclei in pachytene.** (A) Fluorescence micrographs of pachytene-arrested EW104 cells after 6 hours in SM. SCs are marked by Zip1-GFP. Phalloidin-stained F-actin does not co-localize with DAPI-stained chromatin or Zip1-GFP. (B) Confocal slices of pachytene-arrested LGY0069 cells, which express Zip1-GFP (green) and Lifeact-mCherry (magenta). (C) Maximum-intensity projections through the nuclei of LGY0069 cells.
(TIF)

**S6 Fig. MTH assembly is dependent on F-actin polymerization.** Fluorescence microscopy of (A) DMSO-treated and (B) Latrunculin A-treated EW104 (*ndt80Δ*) pachytene cells. (C) Projection cryo-EM image of a DMSO-treated EW104 cell, showing MTH bundles inside the nucleus. One set of MTH bundles is enclosed in the dashed oval. (D) Projection cryo-EM image of a Latrunculin A-treated EW104 *ndt80Δ* cell. In panels C and D, the linear pattern running along the diagonal form the upper left to the lower right is from knife marks. In panel C, the wavy patterns running along the 2 o'clock to 8 o'clock diagonal are from cryosection crevassing. These image features are not devitrification artifacts; they are absent from the tomographic slices (computational) in other figures because they can be computationally excluded. 15 of 24 cryosections of DMSO-treated cells contain MTH bundles while 0 out of 66 cryosections of Lat-A-treated cells contain MTH bundles. All cells were incubated in SM for 6 hours prior to drug treatment. See also Table 2.
(TIF)

**S7 Fig. The cryotomograms have the correct handedness.** (A) Density map of the 80S yeast ribosome [97], simulated at 35 Å resolution with the wrong handedness (mirrored) and correct one. The "beak" motif is indicated by the arrow. (B) Subtomogram average of cytoplasmic ribosomes from *ndt80Δ* cells. (C) Fourier shell correlation (FSC) plot of the subtomogram average in panel B. The resolution is ~ 50 Å, based on the "Gold standard" FSC = 0.5 criterion.
(TIF)

**S8 Fig. MTH bundles are extensive.** Volta cryotomographic slices (computational) of six sequential cryosections of a single *ndt80Δ* cell, isolated after 8 hours in SM. The schematic (not to scale) shows the relationship between the numbered cryosections and the cytology. Major features are the cell nucleus (blue circle) and the MTH bundles (green blob). The MTHs are in the center of the nucleus while the spindle is anchored at the lower left, with many nuclear microtubules visible in cryosections 4, 5, and 6.
(TIF)

**S1 Movie. *In situ* organization of the yeast MTH bundle.** This movie shows a representative MTH bundle in a WT cell (NKY611), a 3-D model of the MTH bundle, and a subtomogram average of an MTH segment.
(MP4)

**S1 Table. Strains used.**
(DOCX)

**S2 Table. Primers, 5' → 3'.**
(DOCX)

**S3 Table. Cryo-ET details.**
(DOCX)

**S4 Table. Cryo-EM details.**
(DOCX)

**S5 Table. Template-matching details.**
(DOCX)

# Acknowledgments

We thank the CBIS microscopy staff for support and training. We thank Soni Lacefield, David Kaback, Tadasu Nozaki, and Nancy Kleckner for sharing strains and for their advice. We thank Mikhail Eltsov for recognizing that the MTH bundles resemble a cholesteric phase and Akira Shinohara for sharing results prior to publication. We acknowledge Diamond Light Source for access and support of the cryo-EM facilities at the UK's national Electron Bio-imaging Centre (eBIC) [under proposal BI23297].

# Author Contributions

**Conceptualization:** Olivia X. Ma, Lu Gan.

**Data curation:** Olivia X. Ma, Lu Gan.

**Formal analysis:** Olivia X. Ma, Lu Gan.

**Funding acquisition:** Lu Gan.

**Investigation:** Olivia X. Ma, Wen Guan Chong, Joy K. E. Lee, C. Alistair Siebert, Andrew Howe, Lu Gan.

**Methodology:** Olivia X. Ma.

**Project administration:** Lu Gan.

**Supervision:** Shujun Cai, Jian Shi, Lu Gan.

**Visualization:** Olivia X. Ma, Lu Gan.

**Writing – original draft:** Olivia X. Ma, Lu Gan.

**Writing – review & editing:** Olivia X. Ma, Wen Guan Chong, Joy K. E. Lee, C. Alistair Siebert, Andrew Howe, Peijun Zhang, Uttam Surana, Lu Gan.

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
