## [Decision Letter · Decision Letter 0]

21 Sep 2021

PONE-D-21-26731Cryo-ET detects bundled triple helices but not ladders in meiotic budding yeastPLOS ONE

Dear Dr. Gan,

Thank you for submitting your manuscript to PLOS ONE. After careful consideration, we feel that it has merit but does not fully meet PLOS ONE’s publication criteria as it currently stands. Therefore, we invite you to submit a revised version of the manuscript that addresses the points raised during the review process.

The one of the reviewers has noted that you have not indicated how you are going to make all your data publicly available.  It is the policy of the PLOS journals that all data associated with the manuscript is available.  Typically this means placing that data in a public repository such as Dryad. As editor, I think many of the criticisms of your manuscript can be addressed through scaling back your assertions and some follow up analysis so I do encourage a revised version to be submitted.

We look forward to receiving your revised manuscript.

Kind regards,

Jennifer C. Fung

Academic Editor

PLOS ONE

Journal Requirements:

[We thank the CBIS microscopy staff for support and training. We thank Soni Lacefield, David Kaback, Tadasu Nozaki, and Nancy Kleckner for sharing strains and for their advice. We thank Mikhail Eltsov for recognizing that the MTH bundles resemble a cholesteric phase and Akira Shinohara for sharing results prior to publication. We acknowledge Diamond Light Source for access and support of the cryo-EM facilities at the UK's national Electron Bio-imaging Centre (eBIC) [under proposal BI23297], funded by the Wellcome Trust, MRC and BBRSC. US was funded by the Biomedical Research Council of A*STAR (Agency for Science, Technology and Research), Singapore. OXM, SC, WGC, JKEL, and LG were supported by a Singapore Ministry of Education Tier 1 grants R-154-000-A49-114 and R-154-000-B42-114 and Tier 2 grant MOE2019-T2-2-045.]

 [Ministry of Education - Singapore (MOE):Peijun Zhang,Jian Shi,Uttam Surana R-154-000-A49-114; Ministry of Education - Singapore (MOE):Olivia X. Ma,Wen Guan Chong R-154-000-B42-114; Ministry of Education - Singapore (MOE):Olivia X. Ma,Wen Guan Chong MOE2019-T2-2-045; A*STAR | Biomedical Research Council (BMRC):Uttam Surana; Wellcome Trust:Alistair Siebert; UKRI | Medical Research Council (MRC):Alistair Siebert; BBRSC:Alistair Siebert]

 [Ministry of Education - Singapore (MOE):Peijun Zhang,Jian Shi,Uttam Surana R-154-000-A49-114; Ministry of Education - Singapore (MOE):Olivia X. Ma,Wen Guan Chong R-154-000-B42-114; Ministry of Education - Singapore (MOE):Olivia X. Ma,Wen Guan Chong MOE2019-T2-2-045; A*STAR | Biomedical Research Council (BMRC):Uttam Surana; Wellcome Trust:Alistair Siebert; UKRI | Medical Research Council (MRC):Alistair Siebert; BBRSC:Alistair Siebert].  

Please state what role the funders took in the study. If the funders had no role, please state: "The funders had no role in study design, data collection and analysis, decision to publish, or preparation of the manuscript.

Reviewers' comments:

Reviewer's Responses to Questions

**Comments to the Author**

1. Is the manuscript technically sound, and do the data support the conclusions?

Reviewer #1: Partly

Reviewer #2: Yes

2. Has the statistical analysis been performed appropriately and rigorously? 

Reviewer #1: N/A

Reviewer #2: Yes

3. Have the authors made all data underlying the findings in their manuscript fully available?

Reviewer #1: No

Reviewer #2: Yes

4. Is the manuscript presented in an intelligible fashion and written in standard English?

Reviewer #1: Yes

Reviewer #2: Yes

5. Review Comments to the Author

Reviewer #1: In this manuscript, Ma et al report a cryo-electron tomography analysis of meiotic budding yeast that reveal a number of regular structures that they have named ‘meiotic triple helices’ (MTHs). The structures the authors observe are intriguing and to my knowledge novel. The key question of course is what do these structures represent – which proteins make up these structures and what is their function in meiosis? The authors do not address these questions. Nevertheless, their observation and tomographical description constitute novel findings that are suitable for publication in PLOS One. Whilst I am happy to recommend that the main findings of the study should be published, I have substantial concerns regarding the text and conclusions of the manuscript that must be addressed. Hence, I recommend that the manuscript is re-evaluated following major revisions.

Major concerns:

I do not understand why the text focusses so heavily on the synaptonemal complex (SC) when they provide no data on the SC. The authors do demonstrate that the MTHs form in absence of Zip1 or Red1, so are clearly not formed of SC proteins and clearly represent distinct (albeit unknown) structures. Despite this, the title, abstract and substantial parts of the text refer to the SC and appear to imply that the SC’s established structure is incorrect owing to them having observed these distinct structures and not having visualised SCs. I note that they have included some caveats in the text but the overall implication is clear and is misleading. Their argument is that they did not visualise SC structure so it must be absent or different to its established structure. This is not a reasonable argument – not visualising an SC does not mean that it is not there as there are many trivial explanations for it not having been observed. If they want to make this argument, they would have to perform a much more detailed investigation such as by immuno-localising SC components. The authors should re-write the manuscript to focus on their findings – the observation of MTHs – and not on the lack of SCs. This particularly applies to the title (lack of ladders is not a justifiable finding), abstract, section entitled ‘ladder-like motifs are absent’ and the discussion. Also, the authors include an unusual statement in the discussion – “Our data suggests that yeast SCs are not crystalline because the subunits of crystalline materials contact multiple other subunits, i.e., they are densely packed.” This statement is not justified as they have no data on the SC. Also, the logic is flawed as crystals are not necessarily densely packed (in contrast, crystal contacts often lead to low density owing to large solvent channels), and the SC is not a crystalline structure (regularity and polymeric assembly are not sufficient qualities).

I do not understand why the authors have called the novel structures ‘meiotic triple helices’. It is fair to say that they are meiotic, but what is the evidence that they are triple helices? The subtomogram averaging suggests a right-handed twist, but I see no clear evidence for their constitution. The name is slightly ambiguous – do the authors mean that they are formed of three alpha-helices (as in a three-helical coiled-coil) or do they mean that the overall structure is helical with some internal three-fold symmetry? This should be clarified. A width of 12 nm is around 10x the size of a three-helical coiled-coil, so I assume they mean the latter. To say that it is helical requires them to demonstrate a periodicity along the longitudinal axis with a regular twist. The models shown are very short so I think it would be impossible to make these conclusions from these data. Also, what is the evidence that they are ‘triple’ helices? – individual cross-sections are insufficient and this should be supported by computational analysis of density correlations around the longitudinal axis.

What is the evidence that MTHs form crystalline bundles? The images in figure 6 suggest that they form bundles, but not that they are necessarily crystalline. I see that they have included Fourier transforms that demonstrate order in certain directions – this is observed for many bundled structures and are not diagnostic of crystalline behaviour (muscle fibres and hair demonstrate similar features upon X-ray analysis – are these crystals?). The authors should review the definition of a crystal, and consider what differences they would expect to observe between crystalline and non-crystalline bundles, and then re-consider their interpretation.

Reviewer #2: The manuscript "Cryo-ET detects bundled triple helices but not ladders in meiotic budding yeast" by Ma and colleagues presents an interesting set of data showing a relatively novel structure in the nuclei of meiotic yeast cells. Meiosis is an important two-stage cell division program in eukaryotes, which divides the genome in half in preparation for sexual reproduction. A key structural element of meiotic chromosomes is the synaptonemal complex (SC), a ladder-like protein assembly that holds homologs together during their recombination, and plays key signaling roles.

With the recent advances in cryo-electron microscopy and especially cryo-electron tomography (cryo-ET), researchers in the meiosis field have been interested in revisiting ultrastuctural studies of meiotic chromosomes, which were originally performed decades ago using sectioned and stained samples. This study reveals that when examined by cryo-ET, the SC is surprisingly not visible, likely because it is composed of loosely-packed alpha-helical coiled coils and does not provide enough contrast for visibility in cryo-ET. However, this study reveals an unexpected prominent structure termed meiotic triple helices, or MTHs.

Overall, the evidence shown here is clear, the experiments are sound, and the conclusions are warranted. The authors show convincingly that MTHs appear with similar timing as SCs, but are not comprised of SC proteins. Analysis of the tomograms and sub-tomogram averages reveals the MTH structure as a 12-nm wide right-handed triple helix, packed into regular arrays in the cells. These arrays strongly resemble those from a recent report from Shinohara and colleagues (referenced in this report) that proposed that these filaments are made of actin. The earlier study showed that these structures label with immunogold anti-actin antibodies in cryo-EM, and the current study shows that the actin-depolymerizing drug Latrunculin A dissolves the MTHs. Thus, despite the fact that F-actin filaments are double-helical and the MTHs are triple-helical, the balance of evidence suggests that the MTHs are made of actin, albeit possibly a non-canonical filamentous form. This is interesting, and definitely worth publication.

I have a few minor comments/questions that the authors should address before publication. First, I dispute the authors' assertion that one expected structure in these nuclei is highly ordered nucleosome arrays: there is no reason to expect that chromatin should be highly ordered at the nucleosome level simply because of the established chromatin loop-axis model of meiosis. I suggest the authors remove or rephrase these assertions. Second, line 312 seems to be an incomplete section? Please address. Finally, given that the balance of evidence suggests that these are non-canonical actin structures, I suggest that the authors spend some time (and possibly a new figure) comparing the known structure of F-actin to that of the MTHs. Figure 5 reveals not only the width and triple-helical nature of the MTHs, but also the rough subunit spacing along the filament. How do these measurements compare to F-actin filaments? Is there any precedent for an actin triple helix? Finally, how does the spacing and packing of MTHs compare to known F-actin filament bundles?

6. PLOS authors have the option to publish the peer review history of their article (what does this mean?). If published, this will include your full peer review and any attached files.

Reviewer #1: No

Reviewer #2: No

---

## [Author Response · Author response to Decision Letter 0]

31 Jan 2022

<< Note that the rebuttal .docx file is easier to read because I have color-coded and bolded our responeses >>

In the rebuttal below, the Reviewer comments are reproduced in black regular font. Our rebuttal is in blue bold font.

5. Review Comments to the Author

We thank the Reviewers for taking their time to give critical feedback on our manuscript. The version reviewed here at PLOS One had already been revised in light of the Review Commons reviews. Notably, we had toned down nearly every claim regarding the synaptonemal complex (SC). The new reviews here recommend three major changes:

1. Tone down / limit our discussion on the SCs even more.

2. Clarify what we mean by MTH.

3. Add more discussion on the relationship between the MTH and actin polymers.

We have addressed all these comments in the rebuttal that follows. We have also updated the Materials and Methods on the conditions for time course experiments: in our efforts to make an improved Figure 1, we found that when we used the original protocol, we could not pellet enough cells. After a few months of trial and error, we figured out that we have to prepare much-larger volumes of sporulating cells to enable pelleting.

One of the reviewers has noted that you have not indicated how you are going to make all your data publicly available. It is the policy of the PLOS journals that all data associated with the manuscript is available. Typically this means placing that data in a public repository such as Dryad.

We have renamed our “Data sharing” heading to “Data Availability Statement”, following PLOS guidelines. We have also added more details in lines 687-690:

“All of the deposited cryo-EM data are unbinned raw tilt series, meaning that they are faithful copies of the microscope’s original output that have not been degraded in any way. The raw confocal microscopy data are available at the BioImage Archive [99].”

Reviewer #1: In this manuscript, Ma et al report a cryo-electron tomography analysis of meiotic budding yeast that reveal a number of regular structures that they have named ‘meiotic triple helices’ (MTHs). The structures the authors observe are intriguing and to my knowledge novel. The key question of course is what do these structures represent – which proteins make up these structures and what is their function in meiosis? The authors do not address these questions. Nevertheless, their observation and tomographical description constitute novel findings that are suitable for publication in PLOS One. Whilst I am happy to recommend that the main findings of the study should be published, I have substantial concerns regarding the text and conclusions of the manuscript that must be addressed. Hence, I recommend that the manuscript is re-evaluated following major revisions.

Major concerns:

I do not understand why the text focusses so heavily on the synaptonemal complex (SC) when they provide no data on the SC. The authors do demonstrate that the MTHs form in absence of Zip1 or Red1, so are clearly not formed of SC proteins and clearly represent distinct (albeit unknown) structures. Despite this, the title, abstract and substantial parts of the text refer to the SC and appear to imply that the SC’s established structure is incorrect owing to them having observed these distinct structures and not having visualised SCs. I note that they have included some caveats in the text but the overall implication is clear and is misleading. Their argument is that they did not visualise SC structure so it must be absent or different to its established structure.

Our experiments imply that the SC’s structure is different from what’s established; not that there are no SCs. The absence of ladder-like densities in our data of cells that clearly have SCs by fluorescence data does not mean that we provide no data on the SC. We are simply reporting a negative result here. Furthermore, the absence of ladder-like densities are also reported by the published Shinohara lab paper (www.nature.com/articles/s42003-021-02545-9), where they state:

“Unfortunately, we could not efficiently detect SCs in our cryosections, although SCs could be detected by the heavy metals-staining of fixed meiotic cells”

We have reworded the abstract to use purely cytological terminology, which rigorously describes our data: no ladder-like complexes of ordered chromatin loop-like structures were detected:

“In meiosis, cells undergo two sequential rounds of cell division, termed meiosis I and meiosis II. Textbook models of the meiosis I substage called pachytene show that nuclei have conspicuous 100-nm-wide, ladder-like synaptonemal complexes and ordered chromatin loops. It remains unknown if these cells have any other large, meiosis-related intranuclear structures. Here we present cryo-ET analysis of frozen-hydrated budding yeast cells before, during, and after pachytene. We found no cryo-ET densities that resemble dense ladder-like structures or ordered chromatin loops. Instead, we found large numbers of 12-nm-wide triple-helices that pack into ordered bundles. These structures, herein called meiotic triple helices (MTHs), are present in meiotic cells, but not in interphase cells. MTHs are enriched in the nucleus but not enriched in the cytoplasm. Bundles of MTHs form at the same timeframe as synaptonemal complexes (SCs) in wild-type cells and in mutant cells that are unable to form SCs. These results suggest that in yeast, SCs coexist with previously unreported large, ordered assemblies.”

Please see our further details below.

This is not a reasonable argument – not visualising an SC does not mean that it is not there as there are many trivial explanations for it not having been observed. If they want to make this argument, they would have to perform a much more detailed investigation such as by immuno-localising SC components.

Our argument would be unreasonable if a tiny minority of cells contained SCs, but we solved this issue by using strains that either arrest in pachytene or can synchronously enter pachytene. And we have provided fluorescence data that nearly every cell contains SCs. The SCs are pervasive throughout the nucleus, so it is unlikely that we have missed the SC in every single cryotomogram. As seen in the textbook model from Byers and Goetsch “Electron microscopic observations on the meiotic karyotype of diploid and tetraploid Saccharomyces cerevisiae”, the SCs span span the entire nucleus so nearly every cryosection cut through a pachytene nucleus would sample a few segments of the SC:

Fig 2 from Byers and Goetsch 1975, PNAS. Continuous lines = lateral elements. Dashed lines = central elements. The dotted pattern is the nucleolus. The set of SCs is bounded by the nuclear envelope.

We did not imply that there is no SC as we have used the presence of SCs, as visualized by fluorescence microscopy of GFP-tagged Zip1, to ascertain that we had arrested cells in pachytene. To make it even more unmistakable that we are not implying there are no SCs, we have added the following sentence as lines 346-348:

“Our data do not imply – and have not been interpreted – as evidence of the absence of SCs. The SCs are present in the cells, based on the presence of Zip1-GFP signals. ”

Also note that it is not possible to do immunolocalization in cryo-ET samples because the samples have to be maintained at temperatures lower than −135°C.

The authors should re-write the manuscript to focus on their findings – the observation of MTHs – and not on the lack of SCs. This particularly applies to the title (lack of ladders is not a justifiable finding), abstract, section entitled ‘ladder-like motifs are absent’ and the discussion.

Our revised title and the similar-sounding section heading are correct – we have indeed not observed ladder-like motifs. This would be the first motif any student of meiosis will look for in a cryotomogram of pachytene cells. It is also what people mean when they ask us “do you see the SC?” given how the ladder-like motifs are the most prominent ultrastructural features expected of such cells. Furthermore, the absence of ladder-like motifs has been reproduced by the Shinohara lab as stated above, though they explicitly use the term “SC” instead of ladder-like motif.

Note that the Shinohara lab did not use cryosections, at least as defined by the cryo-EM community. In cryo-EM, cryosections are cut at cryogenic temperatures from frozen-hydrated samples that have never been warmed beyond −135°C; and are also imaged at temperatures colder than −135°C. Instead they used two types of samples: (1) plastic sections of either “rapid-frozen” samples and (2) high-pressure-frozen, freeze-substituted samples. Both of these types of samples are heavy-metal stained and preserve cellular ultrastructure better than the forms of traditional EM that were used to visualize ladders in the historical studies in the 1970s.

Also, the authors include an unusual statement in the discussion – “Our data suggests that yeast SCs are not crystalline because the subunits of crystalline materials contact multiple other subunits, i.e., they are densely packed.” This statement is not justified as they have no data on the SC. 

Our original sentence is poorly worded because it sounded like we had identified densities belonging to the SC. To do so would have required localization precision well beyond what is currently feasible given how crowded the nucleus is, in the same vein that it is not yet possible to identify where the Shinohara group’s anti-actin immuno-EM signals come from (see our response on this matter to Reviewer 2 below). We made our claim based on the fact that SCs are broadly distributed throughout the nucleus and have dimensions of ~ 100 nm. These properties make it unlikely that SCs were missed in every single one of our tomograms (see Byers & Goetchs’ figure above). We have rewritten this section (lines 351-356) to better explain how our study supports a non-crystalline arrangement of SCs:

“We did not observe any structures that resemble what has been depicted of SCs in the literature. Namely, we did not observe any structure that appears packed to crystalline density, except for the MTH bundles (see below). Note that we are not implying that the SCs are built from MTHs because these are clearly distinct structures as mutants that cannot assemble SCs are still able to make MTHs.”

Also, the logic is flawed as crystals are not necessarily densely packed (in contrast, crystal contacts often lead to low density owing to large solvent channels), and the SC is not a crystalline structure (regularity and polymeric assembly are not sufficient qualities).

Protein crystals indeed have solvent channels, which have been historically exploited for substrate/metal-ion “soaking” experiments, but these channels are not big enough to allow particles the size of nucleosomes to pass. Regarding the term “crystalline”, please see our response to the last comment “What is the evidence that MTHs form crystalline bundles?”, regarding the semantics.

Based on the existing literature, we used to believe that the SC was crystalline, or at least ordered enough to produce layer lines in power spectra. This belief, which is clearly wrong, contributed to our initially wrong conclusion that the MTH bundles were part of the SC. We now appreciate that the SC – at least in yeast – is not crystalline. Unfortunately, the meiosis literature still does not have consensus as to whether the SC is crystalline or not. Figures from textbooks, reviews, and papers almost unanimously depict SCs as crystalline, both in the sense that they are ordered and in the sense that the subunits pack closely. For example, an influential paper in the field ascribes crystalline (or that liquid crystalline) properties to the SC (including yeast):

“The synaptonemal complex has liquid crystalline properties and spatially regulates meiotic recombination factors” – https://elifesciences.org/articles/21455

Below, we reproduce Fig. 6C from that paper:

Note that we are not singling out this paper for its portrayal of SCs, as there are other examples that depict highly ordered SC subunits. We also have not found a paper that deliberately illustrates the SCs or the DNA loops (see below) as irregular, which would be more consistent with our cryo-ET data.

The Reviewer, like us, belongs to the camp that does not think the SC is crystalline. To discuss this issue, we have added an explanation that there is no consensus on whether the SC is crystalline or not in lines 356-357:

“Also note that there is no consensus on whether the SC is crystalline or not, as seen in papers that depict SCs with [55, 65] and without [69, 70] order.”

I do not understand why the authors have called the novel structures ‘meiotic triple helices’. It is fair to say that they are meiotic, but what is the evidence that they are triple helices? The subtomogram averaging suggests a right-handed twist, but I see no clear evidence for their constitution. The name is slightly ambiguous – do the authors mean that they are formed of three alpha-helices (as in a three-helical coiled-coil) or do they mean that the overall structure is helical with some internal three-fold symmetry? This should be clarified.

At the present resolution, the term triple helix is the most compact way to describe these structures. For example, if we were to call the structure a meiotic trilobe polymer, we would be implying that the subunits have 3 lobes. We have added additional explanations to what we mean by helix, in particular that the individual strands are not alpha helices, in lines 123-126:

“The term MTH does not imply that they are composed of three alpha helices (like a three-helix coiled coil). Also, the term does not imply knowledge of the structure’s detailed subunit organization or helical parameters.”

A width of 12 nm is around 10x the size of a three-helical coiled-coil, so I assume they mean the latter. To say that it is helical requires them to demonstrate a periodicity along the longitudinal axis with a regular twist. The models shown are very short so I think it would be impossible to make these conclusions from these data. Also, what is the evidence that they are ‘triple’ helices? – individual cross-sections are insufficient and this should be supported by computational analysis of density correlations around the longitudinal axis.

No, we did not mean that the MTHs are three helical coiled-coils, which we don’t think would even be visible in the tomograms. We have clarified this point in lines 369-370:

“These strands are approximately five times thicker than alpha helices (~ 1 nm) so the MTH is not a three-helix coiled coil.”

To explain that the term MTH is a morphological one and that it can be revised with better data, we added this caveat to the lines 126-128:

“MTH is a morphological term that is limited by the current data and the extent of analysis possible. This term may be revisited after the periodicity and twist along the longitudinal axis are better measured.”

Please also see our related response about the limitations of the current data to Reviewer 2’s comment “Finally, given that the balance of evidence…”.

What is the evidence that MTHs form crystalline bundles? The images in figure 6 suggest that they form bundles, but not that they are necessarily crystalline. I see that they have included Fourier transforms that demonstrate order in certain directions – this is observed for many bundled structures and are not diagnostic of crystalline behaviour (muscle fibres and hair demonstrate similar features upon X-ray analysis – are these crystals?). The authors should review the definition of a crystal, and consider what differences they would expect to observe between crystalline and non-crystalline bundles, and then re-consider their interpretation.

We thank the Reviewer for noting our improper usage of “crystalline”. The Merriam-Webster dictionary has 3 definitions (https://www.merriam-webster.com/dictionary/crystalline):

1: resembling crystal: such as

a: strikingly clear or sparkling

b: CLEAR-CUT

2: made of crystal : composed of crystals

3: constituting or relating to a crystal

Whereas X-ray crystallographers use the second definition as it relates to 3D crystals, cell biologists mostly use the broader third definition. In the context of SCs, the meiosis field uses crystalline to imply that the model of the SC has some properties (dense packing and order) that resemble a 3D crystal. At least one group in the field (Dernburg) writes that the SC resembles a liquid crystal, which has some – but not all – properties of a 3D crystal. We used the term crystalline to invoke the combination of density and order, but we should really treat these two properties separately. We have therefore substituted the terms “ordered” when referring to the parallel packing of the MTHs or to their Fourier power spectral features. In instances where we refer to density, we keep the term crystalline and add the term “density” when needed. In the passages on the SC, we use the term crystalline when referring to the terminology used by others in the field, with the disclaimer that there is no agreement (see above).

Reviewer #2: The manuscript "Cryo-ET detects bundled triple helices but not ladders in meiotic budding yeast" by Ma and colleagues presents an interesting set of data showing a relatively novel structure in the nuclei of meiotic yeast cells. Meiosis is an important two-stage cell division program in eukaryotes, which divides the genome in half in preparation for sexual reproduction. A key structural element of meiotic chromosomes is the synaptonemal complex (SC), a ladder-like protein assembly that holds homologs together during their recombination, and plays key signaling roles.

With the recent advances in cryo-electron microscopy and especially cryo-electron tomography (cryo-ET), researchers in the meiosis field have been interested in revisiting ultrastuctural studies of meiotic chromosomes, which were originally performed decades ago using sectioned and stained samples. This study reveals that when examined by cryo-ET, the SC is surprisingly not visible, likely because it is composed of loosely-packed alpha-helical coiled coils and does not provide enough contrast for visibility in cryo-ET. However, this study reveals an unexpected prominent structure termed meiotic triple helices, or MTHs.

Overall, the evidence shown here is clear, the experiments are sound, and the conclusions are warranted. The authors show convincingly that MTHs appear with similar timing as SCs, but are not comprised of SC proteins. Analysis of the tomograms and sub-tomogram averages reveals the MTH structure as a 12-nm wide right-handed triple helix, packed into regular arrays in the cells. These arrays strongly resemble those from a recent report from Shinohara and colleagues (referenced in this report) that proposed that these filaments are made of actin. The earlier study showed that these structures label with immunogold anti-actin antibodies in cryo-EM, and the current study shows that the actin-depolymerizing drug Latrunculin A dissolves the MTHs. Thus, despite the fact that F-actin filaments are double-helical and the MTHs are triple-helical, the balance of evidence suggests that the MTHs are made of actin, albeit possibly a non-canonical filamentous form. This is interesting, and definitely worth publication.

We disagree that the balance of evidence (notably the immuno-EM data) suggest that the MTHs are made of actin. First, the labeling density is extremely low, as seen in the Shinohara group’s published paper (www.nature.com/articles/s42003-021-02545-9). Second, immuno-EM data only show the approximate locations of actin, which may be a portion of the MTH or one of the numerous adjacent densities that are visible in our cryo-ET data, but are invisible by immuno-EM due to extraction/dehydration/fixation artifacts. Immuno-EM cannot show that the MTHs are composed of actin because there is up to a ~20 nm delocalization error that arises from the two antibodies (each ~10 nm long). For Immuno-EM to show that MTHs are composed of actin, one would need to see the antibodies themselves, with their FAbs bound to the surface of the MTH. We believe that in situ cryo-ET, which reveals far more macromolecular complexes than traditional EM, will gradually force a rethinking of the standards needed for identification. However, such a critical discussion is not suited for this paper.

Because the Shinohara lab’s paper has generated interest in the possible role of intranuclear non-F-actin in meiotic nuclei, we have expanded our Discussion on F-actin (see the responses below).

I have a few minor comments/questions that the authors should address before publication. First, I dispute the authors' assertion that one expected structure in these nuclei is highly ordered nucleosome arrays: there is no reason to expect that chromatin should be highly ordered at the nucleosome level simply because of the established chromatin loop-axis model of meiosis. I suggest the authors remove or rephrase these assertions.

This comment is similar to Reviewer 1’s comment about how the SC (at least in yeast) is not a crystalline structure. We completely agree that the chromatin is not highly ordered. Our previous cryo-ET work in haploid budding yeast, fission yeast, plankton, and mammalian cells – and the related cryo-EM work by others in the field – have consistently failed to detect highly ordered chromatin in situ. However, textbooks and literature continue to depict long rows of ordered chromatin chains protruding from SCs. Below is a recent example, presented in the structural biology review “Architecture and Dynamics of Meiotic Chromosomes” www.annualreviews.org/doi/abs/10.1146/annurev-genet-071719-020235:

We do not mean to single this review out as it is only one of tens of recent publications that illustrate ordered chromatin loops in meiosis. Our manuscript provides experimental images that do not show any evidence of ordered chromatin. We hope that by pointing this observation out using cryo-ET data, people in the field will reconsider how they depict SC-proximal chromatin.

Second, line 312 seems to be an incomplete section? Please address.

This was a placeholder for control experiments in our attempt to push the resolution of our subtomogram analysis. It has been removed because these efforts will take much more time than we had expected.

Finally, given that the balance of evidence suggests that these are non-canonical actin structures, I suggest that the authors spend some time (and possibly a new figure) comparing the known structure of F-actin to that of the MTHs. Figure 5 reveals not only the width and triple-helical nature of the MTHs, but also the rough subunit spacing along the filament. How do these measurements compare to F-actin filaments?

At the low resolution of the current data, the rise along the helical axis (~ 5 nm) could be accommodated by an actin subunit. But this rise could also be accommodated by tubulin or any number of ~ 50 kDa globular proteins, or dimers of 25 kDa proteins. We would need subnanometer resolution to guess at the fold and therefore rule in/out actin and obtain better helical parameters. We now explain how the comparison of helical parameters between MTHs and F-actin is premature in lines 371-374

“If the approximately 5-nm spacing along the long axis arises from 5-nm-wide globular proteins, then the subunit can be accounted for by many types of proteins, including actin, tubulin, and dimers of smaller proteins. Higher-resolution analysis is needed to define the subunit structure.”

Please see our related response to Reviewer 1’s comment “A width of 12 nm is around 10x…”

Is there any precedent for an actin triple helix?

We have not found any papers that describe an actin triple helix. This is now stated in line 403:

“We are unaware of any precedent for triple helical actin polymers.”

Finally, how does the spacing and packing of MTHs compare to known F-actin filament bundles?

A recent preprint from the Swulius lab features cryotomograms of F-actin bundles inside neural growth cones:

https://www.biorxiv.org/content/10.1101/2021.09.17.460569v2

They measure an average inter-filament spacing of 12 nm, which implies denser packing than MTHs. Unlike with MTHs, we are not aware of F-actin bundles that intersect with each other. We now note this in lines 403-407:

“A recent cryo-ET study of bundled F-actin in situ revealed an average nearest-neighbor spacing of ~ 12.3 nm [72] with filaments as close as 8 nm. While some of the MTHs are packed to ~ 12 nm apart, none were seen 8 nm, possible because MTHs are thicker than F-actin.”

---

## [Decision Letter · Decision Letter 1]

14 Mar 2022

Cryo-ET detects bundled triple helices but not ladders in meiotic budding yeast

PONE-D-21-26731R1

Dear Dr. Gan,

We’re pleased to inform you that your manuscript has been judged scientifically suitable for publication and will be formally accepted for publication once it meets all outstanding technical requirements.

Kind regards,

Jennifer C. Fung

Academic Editor

PLOS ONE

Additional Editor Comments (optional):

Reviewers' comments:

Reviewer's Responses to Questions

**Comments to the Author**

1. If the authors have adequately addressed your comments raised in a previous round of review and you feel that this manuscript is now acceptable for publication, you may indicate that here to bypass the “Comments to the Author” section, enter your conflict of interest statement in the “Confidential to Editor” section, and submit your "Accept" recommendation.

Reviewer #1: All comments have been addressed

Reviewer #2: All comments have been addressed

2. Is the manuscript technically sound, and do the data support the conclusions?

Reviewer #1: Yes

Reviewer #2: Yes

3. Has the statistical analysis been performed appropriately and rigorously? 

Reviewer #1: N/A

Reviewer #2: Yes

4. Have the authors made all data underlying the findings in their manuscript fully available?

Reviewer #1: Yes

Reviewer #2: Yes

5. Is the manuscript presented in an intelligible fashion and written in standard English?

Reviewer #1: Yes

Reviewer #2: Yes

6. Review Comments to the Author

Reviewer #1: I have reviewed the manuscript and the authors have addressed all of the points, and have suitably tempered their arguments and provided additional clarifications and caveats. I am still of the opinion that crystalline is an inappropriate description for the SC, whether ordered or not. Even if the transverse filaments were entirely regular, it would be a one-dimensional crystal of only one molecule thick (i.e. pairs of transverse filaments), which isn't really a crystal. Nevertheless, this is really a semantic point, and the authors have presented the necessary caveats. Overall, I think it is now suitable to be accepted for publication.

Reviewer #2: The authors have done a good job addressing the concerns of both reviewers. While I still disagree with some of the authors' interpretations, this is more of a healthy scientific debate than a fundamental problem with the paper. Hopefully, this paper will inspire some rethinking and debate within the field, driving progress.

7. PLOS authors have the option to publish the peer review history of their article (what does this mean?). If published, this will include your full peer review and any attached files.

Reviewer #1: No

Reviewer #2: No

---

## [Editor Report · Acceptance letter]

30 Mar 2022

PONE-D-21-26731R1 

Cryo-ET detects bundled triple helices but not ladders in meiotic budding yeast 

Dear Dr. Gan:

I'm pleased to inform you that your manuscript has been deemed suitable for publication in PLOS ONE. Congratulations! Your manuscript is now with our production department. 

Kind regards, 

on behalf of

Dr. Jennifer C. Fung 

Academic Editor

PLOS ONE